# ComfyMind: Toward General-Purpose Generation via Tree-Based Planning and Reactive Feedback

**Litao Guo**[*]
HKUST(GZ)

**Xinli Xu**[*]
HKUST(GZ)

**Luozhou Wang**
HKUST(GZ)

**Jiantao Lin**
HKUST(GZ)

**Jinsong Zhou**
HKUST(GZ)

**Zixin Zhang**
HKUST(GZ)

**Bolan Su**
Bytedance

**Ying-Cong Chen**[†]
HKUST(GZ), HKUST

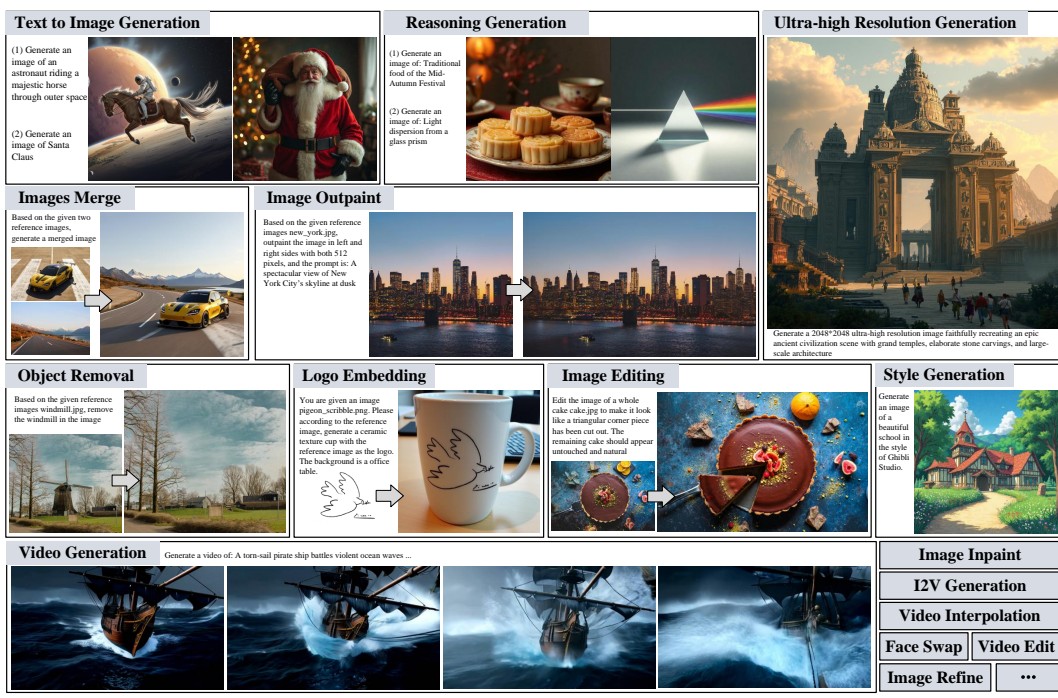

Figure 1: Overview of generative and editing capabilities supported by ComfyMind.

## Abstract

With the rapid advancement of generative models, general-purpose generation has gained increasing attention as a promising approach to unify diverse tasks across modalities within a single system. Despite this progress, existing open-source frameworks often remain fragile and struggle to support complex real-world applications due to the lack of structured workflow planning and execution-level feedback. To address these limitations, we present ComfyMind, a collaborative AI system designed to enable robust and scalable general-purpose generation, built on the ComfyUI platform. ComfyMind introduces two core innovations:

[*]Indicates Equal Contribution
[†]Indicates Corresponding Author

39th Conference on Neural Information Processing Systems (NeurIPS 2025).

Semantic Workflow Interface (SWI) that abstracts low-level node graphs into callable functional modules described in natural language, enabling high-level composition and reducing structural errors; Search Tree Planning mechanism with localized feedback execution, which models generation as a hierarchical decision process and allows adaptive correction at each stage. Together, these components improve the stability and flexibility of complex generative workflows. We evaluate ComfyMind on three public benchmarks: ComfyBench, GenEval, and Reason-Edit, which span generation, editing, and reasoning tasks. Results show that ComfyMind consistently outperforms existing open-source baselines and achieves performance comparable to GPT-Image-1. ComfyMind paves a promising path for the development of open-source general-purpose generative AI systems. Project page: https://github.com/EnVision-Research/ComfyMind

## 1 Introduction

The rapid development of visual generative models has demonstrated remarkable performance across multiple generative tasks, including text-to-image generation [1–3], image editing [4–6], and video generation [7–9]. In recent years, research has gradually shifted towards end-to-end general-purpose generative models [10–13], aiming to handle these diverse tasks within a single unified model. However, existing open-source general-purpose generative models [10–12] still face a series of challenges, including instability in generation quality and a lack of structured planning and composition mechanisms required to handle complex, multi-stage visual workflows. These limitations affect the model's performance and robustness in real-world applications. In contrast, the newly released OpenAI's GPT-Image-1 [13] has garnered widespread attention for its remarkable capabilities in unified image generations [14]. However, the closed-source nature of GPT-Image-1 and its primary focus on image generation tasks limit its applicability and scalability across broader multimodal generation tasks.

The ComfyUI platform provides a potential path toward achieving an open-source general-purpose generative approach. ComfyUI is an open-source platform designed to create and execute generative workflows, offering a node-based interface that allows users to construct visual generative workflows represented as JSON according to their needs. The platform's modular design offers high flexibility in constructing workflows. However, despite its flexibility, building complex workflows from scratch remains a challenge, particularly when dealing with customized or intricate task requirements, which demand substantial expertise and considerable time for trial and error. To address this, recent research [15, 16] has begun exploring the use of large language models (LLMs) to construct customized workflows, thereby enabling general-purpose visual generation.

Building on prior work based on ComfyUI [17], such as ComfyAgent [16], an automated solution for generating workflows from natural language instructions was proposed. ComfyAgent employs a multi-pronged mechanism to convert natural language instructions into executable workflows. This approach involves pseudo-code translation to convert JSON structures into Python-like code, dynamic retrieval [18] of node documentation to standardize outputs, and refinement at the workflow level. While this has made significant progress, ComfyAgent also reveals two core issues in low-level workflow generation. First, it treats workflow construction as a flat, token-based decoding task,

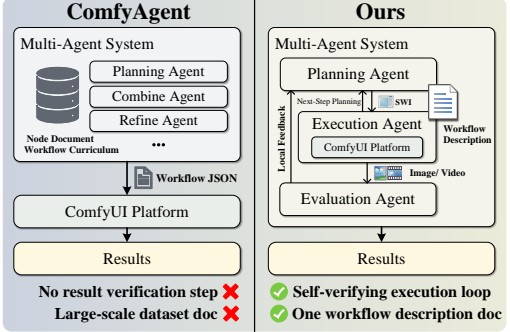

Figure 2: Structural comparison between ours and ComfyAgent.

making it difficult to effectively model modularity and hierarchy. This leads to node omissions, semantic mismatches, and fragile compositions that are challenging to generalize across tasks. Second, the system lacks an execution-level feedback mechanism. Once the workflow is constructed, the system cannot obtain any feedback or error information during generation, hindering incremental corrections and reducing overall robustness.

To address the challenges faced by ComfyAgent, we drew inspiration from human user workflow construction paradigms. We observed that human users typically do not build complex workflows from

scratch but instead decompose tasks into smaller subtasks and select appropriate template workflows for each subtask based on higher-level semantics. This modular, step-by-step planning process, combined with localized feedback strategies, allows for incremental refinement and adaptation. When failure occurs, adjustments are made locally rather than globally. This hierarchical planning and reactive feedback strategy enhances the ability to solve complex problems and increases robustness.

In this work, we simulate this human-like strategy and propose a novel framework called **ComfyMind**. As shown in Figure 1, our framework exhibits strong generality, supporting a wide range of image and video generation and editing tasks. The framework represents workflow generation as the semantic combination of template workflows, rather than the token-based synthesis of node configurations. Specifically, ComfyMind treats template workflows as atomic semantic modules, each with a clearly defined function, input/output interfaces, and natural language descriptions. By reasoning over these higher-level components, ComfyMind achieves more stable and controllable task compositions. In addition, the semantic-level abstraction also enables effortless integration of new workflows, allowing ComfyMind to quickly adapt to emerging community-contributed models and tasks.

ComfyMind consists of two core mechanisms. The first is the **Semantic Workflow Interface (SWI)**, which abstracts low-level node graphs into callable semantic functions annotated with structured inputs, outputs, and natural-language captions. This abstraction allows language models to operate on workflows at the semantic level, reducing exposure to platform-specific syntax and minimizing structural errors. The second mechanism is a **Search Tree Planning with Local Feedback Execution**, which models task execution as a hierarchical decision process. Each node in the planning tree represents a sub-task, and each edge corresponds to a selected SWI module. During execution, failures trigger localized corrections at the current tree layer, avoiding full-chain regeneration and significantly improving robustness. The comparison with previous work is shown in Figure 2.

We evaluate ComfyMind on three public benchmarks: ComfyBench, GenEval, and Reason-Edit. On ComfyBench, ComfyMind improves the workflow pass rate from 56.0% to 100%, and the task resolution rate from 32.5% to 83.0%. On GenEval, it achieves an overall performance score of 0.90, outperforming all baseline methods including GPT-Image-1. On Reason-Edit, it reaches a GPT-score of 0.906, surpassing all open-source systems and matching the performance of proprietary models.

**Our contributions are summarized as follows:**

- We introduce **ComfyMind**, a general-purpose generation framework that conceptualizes visual content creation as a modular, semantically structured planning process, and can be instantiated on node-based execution systems (e.g., ComfyUI) that support modular composition and hierarchical task planning.

- We propose a unified control mechanism that integrates a Semantic Workflow Interface for high-level modular abstraction with Search Tree Planning with Local Feedback Execution, enabling semantically grounded composition, adaptive correction, and robust realization of complex multi-stage workflows.

- We validate ComfyMind on three public benchmarks: ComfyBench, GenEval, and Reason-Edit, covering tasks in generation, editing, and reasoning. The results demonstrate strong performance, broad task generality, and the potential of semantic workflow composition as a foundation for general-purpose visual generation.

## 2 Related work

### 2.1 General-Purpose Visual Generation

Traditional visual generative models typically design task-specific architectures [19–21] to address various generation tasks. With the rapid development of generation models, many studies have shifted towards general-purpose generation models, aiming to solve a wide range of generation tasks using a single model [22–24, 11, 25–29]. These approaches generally unify image understanding and image generation within one model, enabling more efficient and consistent performance across different tasks. Recently, OpenAI's closed-source model GPT-Image-1 [13] has gained significant attention due to its remarkable capabilities in image generation, editing, and modification, achieving top performance across multiple domains.

This study aims to leverage the open-source platform ComfyUI to design a collaborative AI system. By constructing flexible workflows for each task, we strive to achieve general-purpose visual generation. Compared to current general-purpose generation methods, our approach offers distinct advantages: it not only encompasses video modalities but also effectively handles the structured planning and integration of complex, multi-stage visual workflows.

## 2.2 Planning and Feedback Mechanisms in Collaborative AI Systems

In collaborative AI systems, planning and feedback mechanisms are essential for efficiently managing complex tasks. Early systems like AutoGPT [30] generate static goal lists with minimal intermediate feedback, often leading to inefficiencies when sub-goals fail. In contrast, more dynamic systems like AutoGen [31] allow for iterative plan adjustments through dialogue, enhancing adaptability. OpenAgents [32] provides structured execution pipelines tailored to domain-specific tasks but at the cost of flexibility. ReAct [33] introduces a reasoning-acting loop that offers fine-grained control, though it may not be ideal for more complex workflows that require extensive coordination across multiple agents. MetaGPT [34] models human-team roles to standardize collaboration, but it lacks the flexibility needed for dynamic, evolving tasks.

Despite these advancements, existing systems still face challenges in localized error handling and recovery. Many systems lack mechanisms for evaluating intermediate outcomes, making them prone to failure in dynamic, non-linear workflows, such as those encountered in generative tasks like ComfyUI [16]. Motivated by these limitations, we propose a hierarchical search-tree planning strategy combined with local feedback execution to ensure more robust adaptation and fine-grained error recovery during task execution.

# 3 ComfyMind

## 3.1 ComfyUI Platform

ComfyUI is an open-source platform for designing and executing generative pipelines. User-defined workflows are encoded as JSON-based directed acyclic graphs (DAGs) and interpreted node-by-node on the server to produce images or videos. This node-level abstraction significantly lowers the entry barrier for human artists by modularizing the execution process into visual, interpretable components.

ComfyUI, with the efforts of the open-source community, has integrated support for a wide range of advanced models, covering various modalities from image generation to video production. Notable models include Flux [3], Flux Redux [35], ACE++ [36], Segment Anything [37, 38], Hunyuan [8], and Wan [9], among others. This broad support enables users to experiment with state-of-the-art tools for more diverse and sophisticated creative processes. In contrast, building complex workflows from scratch remains a significant challenge. Constructing workflows that meet highly customized or complex task requirements not only requires deep technical knowledge but also demands significant time and trial-and-error, creating considerable barriers for users trying to efficiently develop new workflows or address specific needs.

## 3.2 Semantic Workflow Interface

ComfyAgent aims to construct demand-customized ComfyUI workflows through LLMs to achieve a general approach to visual generation. It has meticulously built a large-scale Node document dataset, designed an intricate collaborative AI system, employed pseudocode to replace JSON for enhanced workspace representation, and utilized RAG technology. However, it still faces issues such as syntax errors in workflows and missing key Nodes, which result in a pass rate of only 56% on ComfyBench. The fundamental reason for this low pass rate lies in the vulnerability of low-level representations to LLMs and the conceptualization of workflow construction as a planar, token-based decoding task, which makes it difficult to effectively model modularity and hierarchy.

In contrast to ComfyAgent's paradigm of building entire workflows at the low-level, we adopt a human-like approach to workflow construction by decomposing generation tasks into modular subtasks, each handled independently by a planning agent. Within each subtask, the planning agent selects the most appropriate atomic workflow from the workflow library as a tool. Unlike complex workflows, each atomic workflow is responsible for a simple, single-step generation process, such

as text-to-image generation or mask generation. In other words, we replace the single token in ComfyAgent with atomic workflows as the minimal unit in workflow construction.

Following this approach, we introduce a semantic workflow interface, which uses natural language functions instead of low-level JSON specifications as an intermediate representation for workflow construction. Each atomic workflow, encapsulating a function, is annotated with a simple natural language description outlining its purpose, required parameters, and usage. Based on this metadata, the planning agent in ComfyMind selects the most appropriate function for invocation. During the invocation, required parameters (e.g., prompts or reference images) and optional high-level constraints are passed. The execution agent then maps the selected function to its corresponding JSON representation, injecting the parameters. Finally, LLMs perform adaptive parameter-level tuning on the JSON to meet additional constraints. The resulting workflow is executed via the ComfyUI platform, thus completing the generation of the individual subtask.

This abstraction allows the LLM to operate entirely at the semantic level, bypassing the complexities of low-level syntactic grammar and the difficulty of effectively modeling modularity and hierarchy. By eliminating this bottleneck, ComfyMind significantly enhances the robustness of execution. The SWI also minimizes the reliance on fine-grained node documentation. While ComfyAgent's operation depends on a meticulously crafted dataset containing 3,205 distinct node descriptions, ComfyMind only requires a single unified document to describe the available atomic workflows. Without the need for RAG, ComfyMind can directly inject workflow metadata into the LLM's context window, ensuring full visibility and eliminating the dependency on external lookups. Ultimately, this reduction in documentation facilitates the seamless integration of newly developed or task-specific workflows. This design enables ComfyMind to quickly integrate emerging workflows from the broader ComfyUI community, while allowing users the flexibility to customize workflow documentation and repositories to meet specific needs.

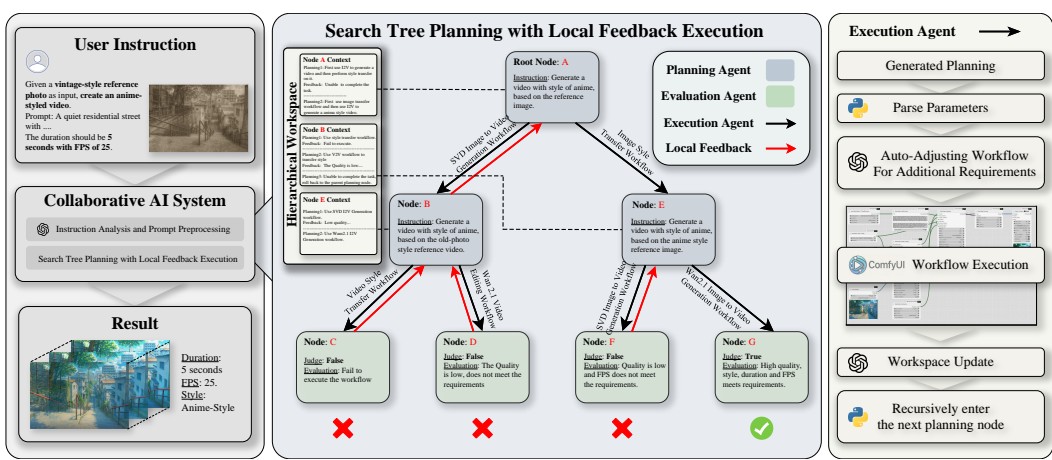

Figure 3: Overview of ComfyMind pipeline. Given a user instruction, the system first parses the task and delegates it to Planning Agent. The Agent incrementally explores a semantic search tree, where each node proposes a candidate workflow and receives local feedback based on execution results.

## 3.3 Search Tree Planning with Local Feedback Execution

As introduced in Section 3.2, SWI enables LLMs to invoke community-validated atomic workflow using natural-language function calls. However, the system must still determine: *how to compose multiple SWI calls into a coherent and task-completing program*. To address this, as sketched in Figure 3, ComfyMind introduces a mechanism we term **Search Tree Planning with Local Feedback Execution**, which formulates workflow construction as a search process over a semantic planning tree. In this structure, each node represents a local planning agent responsible for a specific sub-task, while each edge denotes an execution agent that invokes an SWI function and propagates the result. A complete path from the root to a leaf yields the final visual output that satisfies the user instruction.

At each planning node, the agent examines the current hierarchical workspace state—including text, images, context, and the available workflow documentation. Based on this information, it generates a

chain of SWI functions aimed at advancing the current task. Only the first function in the chain is executed at this stage, with its arguments passed to the execution agent. This transition is equivalent to following an edge in the planning tree.

The execution agent translates the selected function into its canonical JSON form as defined by the SWI, applies lightweight parameter adjustments based on higher-level constraints, and executes the workflow using the ComfyUI platform. Throughout the process, the underlying DAG structure is preserved to ensure syntactic correctness. After execution, a Vision-Language Model (VLM) parses and annotates the generated visual content. The resulting artifact, its semantic description, and the updated task specification collectively define the workspace for the next planning node.

If the planning agent determines that its sub-task can be completed with a single operation, it issues a termination signal and invokes the evaluation agent to assess the final output in terms of semantic alignment and perceptual quality. If the result passes evaluation, the search ends successfully. Otherwise, a failure signal and diagnostic feedback are passed to the parent node, which records the outcome and revises its planning strategy accordingly. If no viable options remain at the current level, the error signal propagates upward. Crucially, all feedback is strictly confined to the current hierarchy level, preventing global rollback and preserving valid partial results.

Compared to the step-by-step observe-then-act execution style of ReAct [39] planners, our method offers complete history tracking and structured backtracking capabilities. This allows the system to roll back only to the most recent viable decision point upon failure, rather than restarting the entire process—thus avoiding redundant recomputation. At the same time, it improves planning stability by preventing repeated re-planning cycles caused by the lack of stable intermediate states, which can otherwise lead to strategy oscillations and failure to converge.

## 4 Experiments

To assess our system's generative capabilities, we conduct a three-pronged evaluation. **Comfy-Bench** [16] quantifies the system's ability to autonomously build workflows and general-purpose generation; **GenEval** [40] evaluates the system's T2I generation capabilities; **Reason-Edit** [6] measures how well complex editing instructions are executed. Experiments demonstrate that our method surpasses the strongest open-source baselines by a substantial margin across all three benchmarks and achieves performance comparable to GPT-Image-1. An ablation study further confirms the contribution of each design component. More experimental results are presented in Appendix A.

### 4.1 Autonomous Workflow Construction

Table 1: Evaluation of Autonomous Workflow Construction on ComfyBench [16].

| Agent | Vanilla | | Complex | | Creative | | Total | |
|---|---|---|---|---|---|---|---|---|
| | %Pass | %Resolve | %Pass | %Resolve | %Pass | %Resolve | %Pass | %Resolve |
| GPT-4o + Zero-shot | 0.0 | 0.0 | 0.0 | 0.0 | 0.0 | 0.0 | 0.0 | 0.0 |
| GPT-4o + Few-shot [41] | 32.0 | 27.0 | 16.7 | 8.3 | 7.5 | 0.0 | 22.5 | 16.0 |
| GPT-4o + CoT [42] | 44.0 | 29.0 | 11.7 | 8.3 | 12.5 | 0.0 | 28.0 | 17.0 |
| GPT-4o + CoT-SC [43] | 45.0 | 34.0 | 11.7 | 5.0 | 15.0 | 0.0 | 29.0 | 18.5 |
| Claude-3.5-Sonnet + RAG [18] | 27.0 | 13.0 | 23.0 | 6.7 | 7.5 | 0.0 | 22.0 | 8.5 |
| Llama-3.1-70B + RAG | 58.0 | 32.0 | 23.0 | 10.0 | 15.0 | 5.0 | 39.0 | 20.0 |
| GPT-4o + RAG | 62.0 | 41.0 | 45.0 | 21.7 | 40.0 | 7.5 | 52.0 | 23.0 |
| o1-mini + RAG | 32.0 | 16.0 | 21.7 | 8.3 | 12.5 | 7.5 | 25.0 | 12.0 |
| o1-preview + RAG | 70.0 | 46.0 | 48.3 | 23.3 | 30.0 | 12.5 | 55.5 | 32.5 |
| ComfyAgent [16] | 67.0 | 46.0 | 48.3 | 21.7 | 40.0 | 15.0 | 56.0 | 32.5 |
| **Ours** | **100.0** | **92.0** | **100.0** | **85.0** | **100.0** | **57.5** | **100.0** | **83.0** |

We assessed the autonomous workflow construction capacity of our method in ComfyBench [16]. ComfyBench comprises 200 graded difficulty generative and editing tasks that span image and video modalities. For each task, the agent must synthesize workflows that can be executed by ComfyUI. The benchmark reports (i) a *pass rate*, reflecting whether the workflow is runable, and (ii) a *resolve rate*, reflecting whether the output satisfies all task requirements.

As shown in Table 1, powered by SWI, our system achieves a **100% pass rate** across all difficulty tiers. Our methods eliminate JSON-level failures that still impede the strongest baseline, ComfyAgent.

More importantly, the proposed Search-Tree Planning with Local-Feedback Execution delivers substantial gains in task resolution: relative to ComfyAgent, **Resolve rate increases by 100%, 292%, and 283% on the Vanilla, Complex, and Creative subsets**, respectively. Appendix further demonstrates that our system successfully addresses a wide spectrum of user instructions. The strong generalization ability and output quality evidenced here point to multi-agent systems based on ComfyUI as a promising avenue toward general-purpose generative AI.

## 4.2 Text-to-Image Generation

### 4.2.1 Quantitative Results

We evaluate our system's capability in T2I generation using GenEval [40]. GenEval measures compositional fidelity across six dimensions, including single or two objects, count, color accuracy, spatial positioning, and attribute binding. We compare our method against three strong categories of baselines: (i) *Frozen Text Encoder Mapping Methods*, represented by SD3; (ii) *LLM/MLLM-enhanced methods*, such as Janus and GoT; and (iii) OpenAI's recently released GPT-Image-1.

As shown in Table 2, our system achieves an overall score of **0.90**, benefiting from its integration of prompt optimization workflows and local feedback execution. This result surpasses all baselines by +0.16 over SD3 and +0.10 over Janus-Pro-7B. Moreover, our system exceeds GPT-Image-1 in five out of six dimensions and the overall score. These results demonstrate that our ComfyUI-based system not only offers strong generality but also is capable of consolidating the strengths of diverse open models, achieving state-of-the-art performance in image synthesis.

Table 2: Evaluation of T2I generation on GenEval [40]. Obj.: Object. Attr.: Attribution.

| Method | Overall | Single Obj. | Two Obj. | Counting | Colors | Position | Attr. Binding |
|---|---|---|---|---|---|---|---|
| *Frozen Text Encoder Mapping Methods* | | | | | | | |
| SDv1.5 [44] | 0.43 | 0.97 | 0.38 | 0.35 | 0.76 | 0.04 | 0.06 |
| SDv2.1 [44] | 0.50 | 0.98 | 0.51 | 0.44 | 0.85 | 0.07 | 0.17 |
| SD-XL [1] | 0.55 | 0.98 | 0.74 | 0.39 | 0.85 | 0.15 | 0.23 |
| DALLE-2 [45] | 0.52 | 0.94 | 0.66 | 0.49 | 0.77 | 0.10 | 0.19 |
| SD3-Medium [2] | 0.74 | 0.99 | 0.94 | 0.72 | 0.89 | 0.33 | 0.60 |
| *LLMs/MLLMs Enhanced Methods* | | | | | | | |
| LlamaGen [46] | 0.32 | 0.71 | 0.34 | 0.21 | 0.58 | 0.07 | 0.04 |
| Chameleon [23] | 0.39 | - | - | - | - | - | - |
| LWM [47] | 0.47 | 0.93 | 0.41 | 0.46 | 0.79 | 0.09 | 0.15 |
| SEED-X [10] | 0.49 | 0.97 | 0.58 | 0.26 | 0.80 | 0.19 | 0.14 |
| Emu3-Gen [24] | 0.54 | 0.98 | 0.71 | 0.34 | 0.81 | 0.17 | 0.21 |
| Janus [11] | 0.61 | 0.97 | 0.68 | 0.30 | 0.84 | 0.46 | 0.42 |
| JanusFlow [22] | 0.63 | 0.97 | 0.59 | 0.45 | 0.83 | 0.53 | 0.42 |
| Janus-Pro-7B [29] | 0.80 | 0.99 | 0.89 | 0.59 | 0.90 | **0.79** | 0.66 |
| GoT [12] | 0.64 | 0.99 | 0.69 | 0.67 | 0.85 | 0.34 | 0.27 |
| GPT-Image-1 [13] | 0.84 | 0.99 | 0.92 | 0.85 | 0.92 | 0.75 | 0.61 |
| *Collaborative AI Systems* | | | | | | | |
| ComfyAgent [16] | 0.32 | 0.69 | 0.30 | 0.33 | 0.50 | 0.04 | 0.04 |
| **Ours** | **0.90** | **1.00** | **1.00** | **1.00** | **0.97** | 0.62 | **0.80** |

### 4.2.2 Qualitative Results

Figure 4 showcases representative and challenging cases from GenEval. Our method follows prompts, outperforming existing models on core constraints like counting, color, position and Attribution Binding. In the counting task, only our system generates exactly four keyboards with clear visual separation. For atypical color and position, we demonstrate superior image quality and instruction consistency. Regarding attribute binding, models like SD3 and Janus-Pro often entangle attributes and fail to localize them correctly. GPT-Image-1, while generally instruction-following, often produces fragmented and visually incoherent compositions. In contrast, our method not only satisfies fine-grained directives but also integrates them into aesthetically coherent, contextually grounded scenes. These qualitative results corroborate the quantitative gains reported earlier.

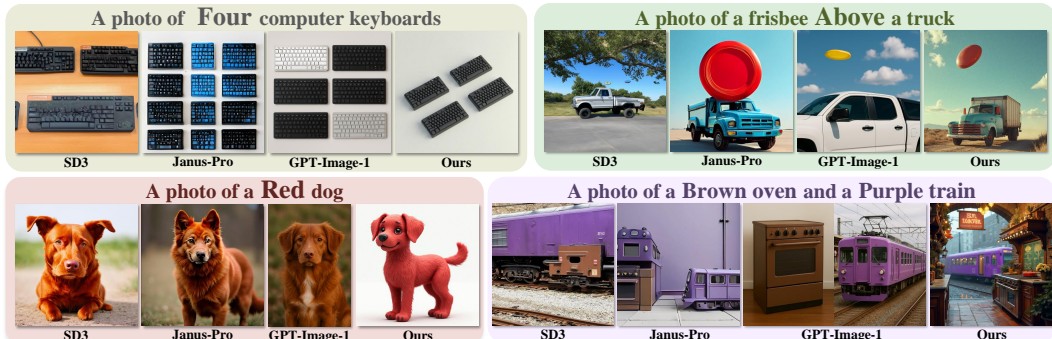

Figure 4: Qualitative comparison on challenging GenEval [40] cases. Under constraints such as counting, color, position and attribute binding, only our method successfully satisfies all instructions, clearly outperforming SD3, Janus-Pro, and GPT-Image-1.

## 4.3  Image Editing

### 4.3.1  Quantitative Results

We further evaluate our system's image editing capability on the Reason-Edit [6]. Following the setting in benchmark, we adopt the *GPT-score* [6] as the evaluation metric. This score quantifies both the semantic fidelity to the editing instruction and the visual consistency of non-edited regions.

We compare our method against the most advanced open-source baselines, including GoT [12], SmartEdit [6], CosXL-Edit [6], SEED-X [10], MGIE [48], MagicBrush [49] and IP2P [4], as well as the strongest closed-source model, GPT-Image-1. As shown in Figure 5, our method achieves a score of **0.906**—the highest among all open-source frameworks. This result represents a substantial improvement of **+0.334** over the previous open-source SOTA SmartEdit (0.572).

Moreover, our method achieves performance competitive with GPT-Image-1 (0.929), narrowing the gap between open and closed models. This gain arises from our system's planning and feedback mechanism, which enables it to synthesise and combine the most effective editing workflows contributed by the ComfyUI community. Through reasoning and iterative correction, our agent can adaptively select diverse workflows, improving the stability and precision of edits across varied scenarios. These results highlight the reasoning-driven editing capability of our system, and suggest strong potential for future performance gains through integration with more powerful workflows and models.

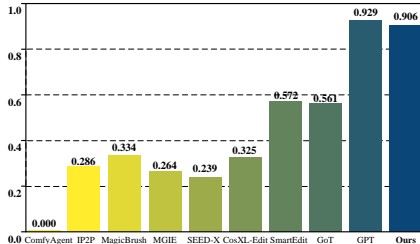

Figure 5: Quantitative Comparison on Reason-Edit [6] benchmark.

### 4.3.2  Qualitative Results

We further present qualitative results to assess the semantic understanding and visual fidelity of our system under challenging editing instructions. We select two representative tasks from the Reason-Edit [6] benchmark. As shown in Figure 6, our method consistently demonstrates the most faithful and visually coherent results across both tasks. Compared to existing open-source baselines, our system not only identifies the correct semantic target (e.g., apple vs. bread vs. orange juice) but also executes edits with minimal disruption to adjacent regions.

Although GPT-Image-1 succeeds in executing editing instructions, it struggles to maintain visual consistency in non-edited regions. As illustrated in Figure 6, GPT-Image-1 loses details in non-edited areas (e.g., patterns on the juice box, yogurt container, and jam jar in the zoom-in views), alters color tones and image style, inaccurately preserves materials (e.g., wood texture), and changes the original aspect ratio. These flaws are also mentioned in GPT-eval [14].

In contrast, our method accomplishes the instructions with minimal edits, effectively preserving visual details, image style, material properties, and proportions. These observations highlight our method's superior capability to perform precise and coherent edits.

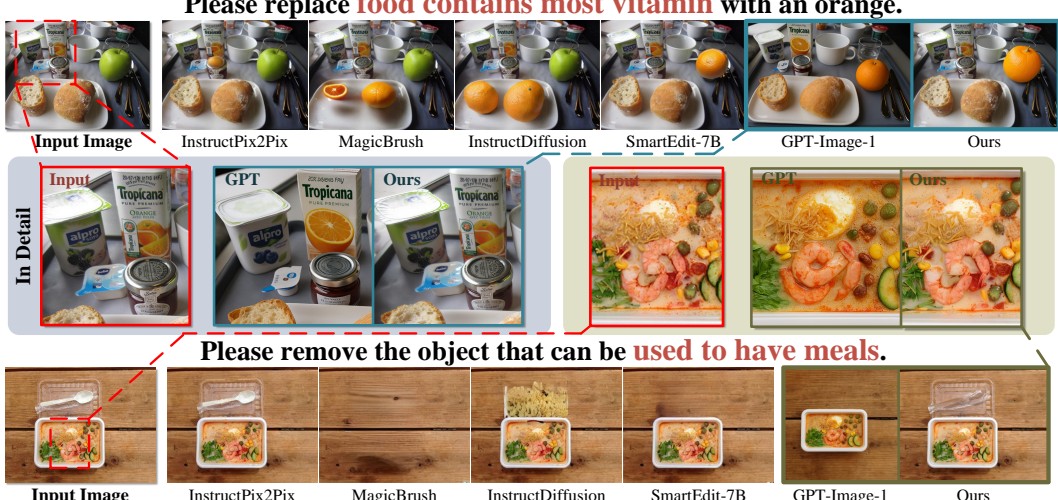

Figure 6: Qualitative Comparison on Reason-edit [6] benchmark.

## 4.4 Ablation Study

To isolate the contributions of our key design components, we conduct an ablation study on the ComfyBench benchmark (Table 3). We evaluate three variants: the full system, a version without search tree planning, and a version without feedback execution. Benefiting from our Semantic Workflow Interface, all variants achieve a 100% pass rate; the main differences lie in the resolve rate.

Table 3: Ablation results of ComfyMind with different architectures on ComfyBench.

| Agent | Vanilla | | Complex | | Creative | | Total | |
|---|---|---|---|---|---|---|---|---|
| | %Pass | %Resolve | %Pass | %Resolve | %Pass | %Resolve | %Pass | %Resolve |
| **Ours** | **100.0** | **92.0** | **100.0** | **85.0** | **100.0** | **57.5** | **100.0** | **83.0** |
| Ours w/o Tree Planning | **100.0** | 86.0 | **100.0** | 43.4 | **100.0** | 50.0 | **100.0** | 66.0 |
| Ours w/o Feedback | **100.0** | 68.0 | **100.0** | 55.0 | **100.0** | 17.5 | **100.0** | 54.5 |

Removing the search tree planning module results in a notable drop in task resolution, particularly on complex tasks (85.0% to 43.4%), underscoring its role in decomposing multi-step instructions and selecting suitable workflows. Similarly, disabling the local feedback mechanism significantly degrades performance, especially on creative tasks (57.5% to 17.5%), highlighting its importance for iterative correction and adaptive refinement. The results confirm that key components are essential for achieving high success rates in autonomous workflow construction.

## 5 Conclusion

In this work, we propose ComfyMind, a novel framework built on the ComfyUI platform that addresses key challenges in general-purpose visual generative AI. By conceptualizing visual content creation as a modular, semantically structured planning process and incorporating tree-based planning with local feedback execution, ComfyMind improves the stability and robustness of multi-stage workflows. Our framework outperforms previous open-source methods and achieves results comparable to GPT-Image-1 on benchmarks ComfyBench, GenEval, and Reason-Edit. ComfyMind offers a promising path towards scalable, open-source solutions for complex generative tasks.

**Discussion** This work focuses on developing a collaborative AI system with general-purpose visual generation capabilities, rather than generating complete workflow JSON files. The goal is to explore

how such a system can plan, execute, and provide feedback for complex tasks through semantic planning, modular composition, and localized corrections. While we use ComfyUI workflow modules for execution, the workflow itself is a component of the system, not the research focus. By operating at a higher semantic level with the SWI, our approach reduces the complexity and uncertainty of workflow generation, offering a more robust and practical solution for real-world applications.

## 6 Acknowledgements

This work is supported by National Natural Science Foundation of China (No. 62206068), HKUST-HKUST(GZ) Cross-Campus Collaborative Research Scheme (Project No. C036) and Guangdong Provincial Department of Science and Technology's '1+1+1' Joint Funding Program for Guangdong-Hong Kong Universities.

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

# A  Supplementary Experiments

## A.1  Supplementary Ablation Study

### A.1.1  Ablation Study on Different LLMs

Table 4: Ablation results of ComfyMind with different LLMs.

| Agent | Vanilla | | Complex | | Creative | | Total | |
|---|---|---|---|---|---|---|---|---|
| | %Pass | %Resolve | %Pass | %Resolve | %Pass | %Resolve | %Pass | %Resolve |
| ComfyMind with GPT-4o | **100.0** | **92.0** | **100.0** | **85.0** | **100.0** | 57.5 | **100.0** | 83.0 |
| ComfyMind with Deepseek-V3 | **100.0** | 90.0 | **100.0** | 71.7 | **100.0** | 60.0 | **100.0** | 78.5 |

To further demonstrate the robustness of our system, we conducted evaluations on the ComfyBench benchmark using different LLMs in ComfyMind. As shown in Table 4, both Deepseek-V3 and GPT-4o achieve strong performance when employed as the primary LLMs. Specifically, both models reached a 100% task pass rate and approximately 80% overall task completion rate. These results further confirm the stability and reliability of our system across different underlying LLMs.

### A.1.2  Ablation Study on Feedback Granularity

To verify the effectiveness of the proposed localized feedback mechanism, we conduct an ablation study on ComfyBench, investigating the impact of different feedback granularity levels on task performance.

We design three feedback granularity levels: *Judgment* contains only binary success/failure signals; *Judgment+Analysis* adds outcome explanations specifying reasons for success or failure; and *Judgment+Analysis+Description (Ours)* includes detailed image-grounded information such as completed progress, visual observations, and suggestions for plan correction.

Three task types (Vanilla, Complex, Creative) are evaluated, with the success rate as the core metric. The ablation results are shown in Table 5.

Table 5: Ablation results of different feedback granularity levels.

| Feedback Type | Vanilla ↑ | Complex ↑ | Creative ↑ | Total ↑ |
|---|---|---|---|---|
| Judgment | 89.0 | 58.33 | 47.5 | 71.5 |
| Judgment+Analysis | 91.0 | 66.67 | 55.0 | 76.5 |
| Judgment+Analysis+Description (Ours) | **92.0** | **85.0** | **57.5** | **83.0** |

As shown in Table 5, task performance steadily improves with increasing feedback granularity. The binary Judgment feedback yields the lowest overall success rate (71.5%), while incorporating analytical explanations (Judgment+Analysis) elevates the total rate to 76.5%. Our full-granularity feedback (Judgment+Analysis+Description) outperforms all variants across task types, particularly in Complex tasks (85.0%, 18.33 pp higher than Judgment), confirming that richer image-grounded feedback provides the planning agent with actionable insights for informed decision-making. Notably, our search tree design mitigates context dilution at each planning node, ensuring fine-grained feedback remains effective without additional computational overhead—validating the rationality of our localized feedback implementation.

## A.2  World knowledge-Informed Semantic Synthesis

To assess our system's capabilities in complex semantic understanding, reasoning, and world knowledge integration for text-to-image generation, we conduct evaluation on the recent WISE [50] benchmark. This benchmark contains three primary categories: cultural commonsense, spatiotemporal reasoning (including Space and Time subcategories), and natural sciences (comprising Physics, Chemistry, and Biology subfields), totaling 25 specialized domains with 1,000 challenging prompts.

The evaluation metric WiScore combines Consistency, Realism, and Aesthetic Quality through weighted normalization, with a maximum score of 1. Higher WiScore indicates stronger capability in accurately depicting objects and concepts using world knowledge. As shown in Table 6, **our method achieves a superior score of 0.85, surpassing all models, including GPT-Image-1 (0.80)**. Our approach significantly enhances world knowledge integration for open-source solutions, outperforming FLUX.1-dev (0.50) by 0.35 points and enabling open-source models to match GPT-Image-1's performance. The exceptional performance on WISE confirms our system's generalizability and high-quality output in generative tasks.

Table 6: Evaluation of World Knowledge-Informed Semantic Synthesis on WISE [50] Benchmark.

| Method | Cultural | Time | Space | Biology | Physics | Chemistry | Overall |
|---|---|---|---|---|---|---|---|
| *Dedicated T2I Models* | | | | | | | |
| FLUX.1-dev [3] | 0.48 | 0.58 | 0.62 | 0.42 | 0.51 | 0.35 | 0.50 |
| FLUX.1-schnell [3] | 0.39 | 0.44 | 0.50 | 0.31 | 0.44 | 0.26 | 0.40 |
| PixArt-Alpha [51] | 0.45 | 0.50 | 0.48 | 0.49 | 0.56 | 0.34 | 0.47 |
| playground-v2.5 [52] | 0.49 | 0.58 | 0.55 | 0.43 | 0.48 | 0.33 | 0.49 |
| SDv1.5 [44] | 0.34 | 0.35 | 0.32 | 0.28 | 0.29 | 0.21 | 0.32 |
| SDv2.1 [44] | 0.30 | 0.38 | 0.35 | 0.33 | 0.34 | 0.21 | 0.32 |
| SD-XL-base-0.9 [1] | 0.43 | 0.48 | 0.47 | 0.44 | 0.45 | 0.27 | 0.43 |
| SD3-Medium [2] | 0.42 | 0.44 | 0.48 | 0.39 | 0.47 | 0.29 | 0.42 |
| SD3.5-Medium [2] | 0.43 | 0.50 | 0.52 | 0.41 | 0.53 | 0.33 | 0.45 |
| SD3.5-Large [2] | 0.44 | 0.50 | 0.58 | 0.44 | 0.52 | 0.31 | 0.46 |
| *Unify MLLM Models* | | | | | | | |
| Emu3 [24] | 0.34 | 0.45 | 0.48 | 0.41 | 0.45 | 0.27 | 0.39 |
| Janus-1.3B [11] | 0.16 | 0.26 | 0.35 | 0.28 | 0.30 | 0.14 | 0.23 |
| JanusFlow-1.3B [22] | 0.13 | 0.26 | 0.28 | 0.20 | 0.19 | 0.11 | 0.18 |
| Janus-Pro-1B [29] | 0.20 | 0.28 | 0.45 | 0.24 | 0.32 | 0.16 | 0.26 |
| Janus-Pro-7B [29] | 0.30 | 0.37 | 0.49 | 0.36 | 0.42 | 0.26 | 0.35 |
| Orthus-7B-base [53] | 0.07 | 0.10 | 0.12 | 0.15 | 0.15 | 0.10 | 0.10 |
| Orthus-7B-instruct [53] | 0.23 | 0.31 | 0.38 | 0.28 | 0.31 | 0.20 | 0.27 |
| show-o-demo [26] | 0.28 | 0.36 | 0.40 | 0.23 | 0.33 | 0.22 | 0.30 |
| show-o-demo-512 [26] | 0.28 | 0.40 | 0.48 | 0.30 | 0.46 | 0.30 | 0.35 |
| vila-u-7b-256 [25] | 0.26 | 0.33 | 0.37 | 0.35 | 0.39 | 0.23 | 0.31 |
| GPT-Image-1 [13] | 0.81 | 0.71 | 0.89 | **0.83** | **0.79** | 0.74 | 0.80 |
| *Collaborative AI Systems* | | | | | | | |
| **Ours** | **0.90** | **0.79** | **0.92** | 0.77 | **0.79** | **0.82** | **0.85** |

## A.3 Quantitative Evaluation for Planning Strategy

To verify the effectiveness of the proposed planning method, we conduct quantitative comparisons with two representative planning strategies on the ComfyBench benchmark.

ReAct is a reactive planning strategy that dynamically adjusts action sequences based on real-time observations without global task decomposition or stable intermediate decision mechanisms. Planning-Act is a canonical multi-agent planning framework that adopts "root-node full planning + recursive subtask execution" and maintains a fixed plan once the plan-act cycle initiates.

We evaluate three core metrics: Resolve Rate, Repetition Rate, and Average Time per Task, which respectively reflect a system's problem-solving capability, stability of exploration, and efficiency. Resolve Rate measures the success rate in solving compositional tasks. Repetition Rate captures the frequency with which atomic workflows are redundantly selected within a single task. Time represents the average execution time per task.

Table 7: Quantitative comparison with baseline methods on ComfyBench.

| Method | Resolve Rate % ↑ | Repetition Rate % ↓ | Time (s) ↓ |
|---|---|---|---|
| ReAct | 54.0 | 34.59 | 257.43 |
| Planning-Act | 66.0 | 28.24 | 279.79 |
| Ours | **83.0** | **8.70** | **225.12** |

As shown in Table 7, our method achieves the best performance across all metrics. ReAct's low Resolve Rate and high Repetition Rate stem from its lack of global planning, leading to inefficient trial-and-error. Planning-Act's rigidity (fixed plan) and redundant workflow selection (absence of stable intermediate decision nodes) result in suboptimal Resolve Rate and the longest execution time. In contrast, our method mitigates strategy oscillations and enhances efficiency by integrating global planning with real-time adaptability and stable intermediate decision nodes, outperforming baselines in both effectiveness and behavioral stability.

## A.4 Supplementary Experiment on Inference Latency

To justify the inference latency trade-off and verify the accompanying performance gains, we conduct supplementary experiments on Geneval, comparing our system with baseline atomic workflows (SD3.5-L, FLUX.1 Dev) that form its foundation. The comparative results of inference time and generation performance are shown in Table 8.

Table 8: Inference latency and performance comparison on Geneval.

| Model | Time (s) ↓ | Overall ↑ | Single Obj. ↑ | Two Obj. ↑ | Counting ↑ | Colors ↑ | Position ↑ | Attr. Binding ↑ |
|---|---|---|---|---|---|---|---|---|
| SD3.5-L | 47.06 | 0.71 | 0.98 | 0.89 | 0.73 | 0.83 | 0.34 | 0.47 |
| FLUX.1 Dev | 56.51 | 0.66 | 0.98 | 0.81 | 0.74 | 0.79 | 0.22 | 0.45 |
| Ours | 119.94 | **0.90** | **1.00** | **1.00** | **1.00** | **0.97** | **0.62** | **0.80** |

Although our system incurs additional time due to planning, evaluation, and possible regeneration, it achieves significantly higher overall scores. The performance gain justifies the roughly 2x increase in runtime, demonstrating that our system pushes the boundary of what current models can accomplish.

# B    Additional Results and Generation Examples

As an extended demonstration, we present ComfyMind's exemplary outputs across diverse generative tasks in Figures 7, 8, and 9, substantiating its versatile adaptability and superior performance in cross-domain generation scenarios.

# C    Implementation Details

## C.1    System Architecture

Logically, the system is divided into two independent subsystems: the ComfyUI server and the automation control system. Both are deployed on the same physical server but operate within isolated runtime environments and listen on separate ports. This design ensures robust separation and scalability in task concurrency handling and resource management. The server is equipped with an NVIDIA RTX A6000 GPU with 48GB of VRAM, providing sufficient computational capacity for generation tasks.

Within the system architecture, the ComfyUI platform is responsible for executing atomic workflows. It is deployed as a local service that listens on a designated port and provides dual interaction interfaces. On one hand, it exposes a web interface through which users can access the frontend via a browser to edit nodes, configure parameters, and perform real-time executions. On the other hand, ComfyUI incorporates both HTTP and WebSocket protocols, supporting standardized RESTful API task submissions and result retrieval, while enabling real-time progress tracking via WebSocket. In this study, the latter interface is primarily utilized. Technically, the client loads an adapted workflow in JSON format, submits a generation request via the HTTP API, and establishes a persistent WebSocket connection to receive real-time execution updates and final results. Built atop the ComfyUI execution layer, ComfyMind functions as the high-level planning and task management system, responsible for process orchestration, scheduling, and result evaluation. It is developed based on the ComfyAgent framework.

## C.2    Compilation of Atomic Workflow Library and Descriptive Documents

To ensure adaptability across multimodal generation and editing tasks, a diverse set of atomic workflows was systematically collected, curated, and organized to form a foundational resource for general-purpose content generation. These workflows were sourced through a combination of selective acquisition and custom development, aiming to balance stability and broad applicability.

First, standard workflows from the ComfyBench benchmark suite were systematically tested. Suboptimal examples were discarded to ensure overall quality. Second, a curated selection of high-quality workflows was extracted from the official ComfyUI website and documentation to meet common generation needs. Additionally, workflows with strong user ratings and proven effectiveness were sourced from popular communities such as Civitai and OpenArt, enhancing the practicality and robustness of the workflow library.

Beyond these standardized resources, functional workflows tailored for complex reasoning tasks were developed by leveraging the flexibility of ComfyMind, in conjunction with LLMs and VLMs. These workflows include capabilities for reason generation, reason editing, and prompt enhancement, supplementing limitations in the standard resource base.

| Text to Image Generation | |
|---|---|
| (1) Generate an image of a cat sitting on a windowsill looking outside |  |
| (2) Generate an image of a small village covered in snow with smoke coming from chimneys |  |
| (3) Generate an image of a hot air balloon floating over a scenic valley at sunrise |  |

| Reasoning Image Generation | |
|---|---|
| (1) Generate an image of Most representative craft of India |  |
| (2) Generate an image of A massive stone statue of a mythical creature that is a prominent historical landmark in Egypt |  |
| (3) Generate an image of Marshmallow over a bonfire |  |

Figure 7: More examples generated by ComfyMind

## Image Editing

| | |
|---|---|
| (1) You are given an image `icecream.jpg`. Please according to the reference image, generate a coffee with ice cream version of the ice cream. |  |
| (2) Based on the given reference images `train.jpg`, replace the train in the image with High-speed train |  |
| (3) Based on the given reference images `dinner.jpg`, remove the knife and fork in the image |  |
| (4) Based on the given reference images `castle.jpg`, replace the castle in the image with Chineses traditional temple |  |
| (5) Generate an image of a `cherries.jpg`, convert the image into a version of advertisement, It is placed on the exhibition stand with advanced lighting and color matching. |  |
| (6) You are given an image `man.jpg`. Generate another photo to show the man as an elderly version of himself. |  |

Figure 8: More examples generated by ComfyMind

## Style Transfer

(1) Based on the given reference image, generate an image describing A little girl is rowing a boat in a downtown park, with a dog on board. In the foreground, willow trees surround the park. In the middle ground, a Ferris wheel and a carousel from the amusement park are visible. In the background, skyscrapers can be seen.

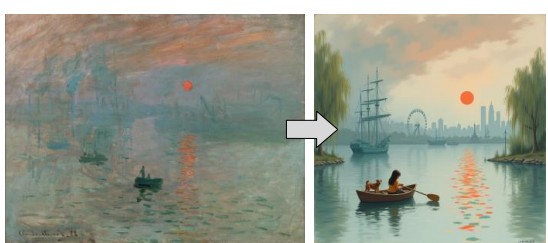

(2) Based on the given masked reference image `tower_mask.png`, generate an image: A tower glowing with lanterns during a traditional night festival, with fireworks lighting up the distant sky and ancient trees surrounding the festive square.

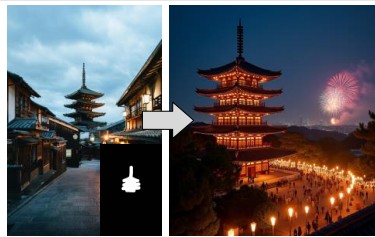

## Image Refine

(3) You are given an portrait image with wrong generated hands`abnormal_hands.png`. Please refine the hands to make them look more natural.

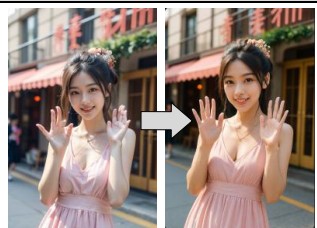

## Image Outpaint

(4) Based on the given reference images `deer.jpg`, outpaint the image in four directions with all 512 pixels, and the prompt is: Standing on the green grass, the deer's body is bathed in a golden glow from the setting sun.

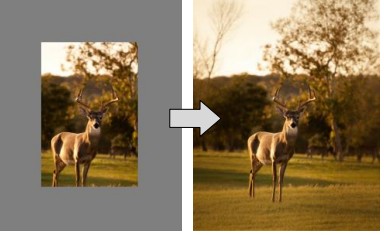

## Image Merge

(5) Based on the given two reference images `pyramid.jpg` and `EiffelTower.jpg`, generate an merged image

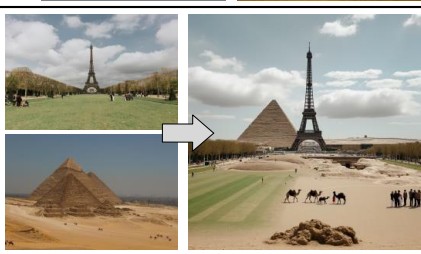

## Face Swap

(6) "Based on the given reference images `portrait.jpg` and `face_reference.jpg`, swap the faces of the two images

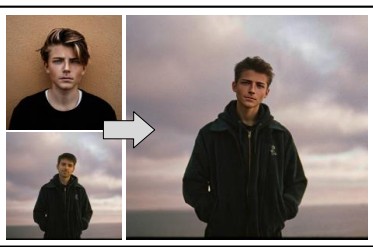

Figure 9: More examples generated by ComfyMind

## C.3 Supplementary Description of the ComfyMind Method

### C.3.1 Semantic Workflow Interface Details

To enable SWI, a unified semantic description schema was developed for all atomic workflows. The system only requires a single, consistent atomic workflow description document to support all reasoning tasks. This document employs a hybrid format combining natural language and standardized parameter interfaces to specify each workflow's core functionality, input/output requirements, and optional extensions.

The core structure includes: the required workflow name, the corresponding prompt and visual input. Optional components include supplementary semantic constraints such as resolution, frame rate, or upscaling ratio. This modular, uniform documentation enables semantic-level workflow selection and parameter matching during reasoning, without requiring parsing of underlying node structures—greatly improving planning efficiency and ease of expansion or maintenance.

### C.3.2 Execution Agent Details

The execution agent in the ComfyMind system plays a crucial role in bridging high-level planning and low-level execution. Once the planning agent completes the decomposition of a subtask and outputs a set of invocation instructions, the execution agent is activated to handle parameter processing, workflow adaptation, execution, and result feedback. The overall logic adheres to the fundamental principle: "if execution succeeds, proceed to the next subtask; if it fails, return to the current node for local backtracking." This ensures tight integration between planning and execution.

Before initiating the execution process, certain specialized semantic workflow interface functions may require a preprocessing step. In such cases, the execution agent invokes LLMs to perform deep reasoning and semantic expansion on the user's input prompt or initial task description. For example, if the user provides a command like "light passes through a prism," the system leverages the LLM's knowledge to infer related phenomena such as spectral dispersion and colorful light bands, thereby optimizing the prompt and providing richer and more accurate input conditions for the subsequent generation process.

Next, the execution agent parses the invocation parameters passed down from the planning agent, loads the corresponding atomic workflow template, and fills in placeholders with information such as the prompt text and input visual data paths. This results in a fully instantiated and executable workflow. To better adapt to the task's specific requirements, the execution agent further adjusts workflow parameters, such as image resolution, sampling steps, and output frame rate. This adaptive tuning is performed at the parameter level without altering the underlying DAG structure of the atomic workflow, thus preserving the correctness of node connections and ensuring execution stability.

Once adaptation is complete, the execution agent submits the workflow to the ComfyUI platform through a standardized interface and establishes a real-time connection to monitor generation progress and status updates. After task execution finishes, the system automatically parses the generated results. It uses a visual-language understanding module to extract semantic descriptions, detailed information, and scene characteristics from the outputs, synchronizing them with the current workspace state. This update provides the most recent context for subsequent subtask reasoning, maintaining continuity and coherence in the reasoning chain.

If the entire process completes successfully, the execution agent returns a success signal and hands off the updated workspace state to the next planning node. If any exceptions occur during execution—such as parameter mismatches, insufficient generation quality, or timeouts—the execution agent returns a detailed failure reason and log information. This allows the planning agent to perform localized search backtracking and re-planning. Through this design, the execution agent effectively ensures seamless integration from high-level semantic instructions to concrete execution, while offering strong adaptability and robust error handling, significantly enhancing the system's overall stability and reasoning flexibility.

### C.3.3 Evaluation Agent Details

The Evaluation Agent is responsible for assessing the quality of generated results and their consistency with the given instructions, ensuring that system outputs meet expected requirements. When the overall process concludes, the system invokes the Evaluation Agent to inspect the generated content against a unified standard and relay the results back to the higher-level planning module.

In its design, the Evaluation Agent adopts the classification-based assessment strategy used by ComfyAgent, dividing workflows according to task types such as text-to-image, image-to-image, text-to-video, image-to-video, and video-to-video, with tailored evaluation criteria for each category. Evaluation is conducted along two main dimensions: the quality of the generation—such as clarity, richness of detail, and visual naturalness—and adherence to instructions, verifying whether the output accurately reflects the user's intended theme, setting, or stylistic directives. Each evaluation returns two components: a binary judgment indicating whether the current task is considered complete, which guides the system on whether to proceed or backtrack; and a detailed failure

analysis when evaluation is unsuccessful, explicitly identifying reasons such as lack of detail, semantic deviation, or stylistic inconsistency, thereby aiding the Planning Agent in refining subsequent inference steps.

To accommodate varying tasks and application requirements, the Evaluation Agent supports dynamic threshold adjustment. The system can raise or lower evaluation standards depending on task complexity, quality expectations, or usage scenarios, enabling a flexible balance between content quality and execution efficiency.

## D    Discussion

GenArtist [54] has made notable advancements in automating complex visual generation leveraging large-scale models. Given the inherent similarity between GenArtist and our work, it is essential to clarify their differences and positioning. Our work and GenArtist share a common vision: addressing the challenges posed by complex instructions via a modular paradigm. A fundamental divergence in objectives exists between the two works: GenArtist focuses on image-centric tasks, whereas our research aims to develop a scalable and intelligent framework tailored for the more general-purpose, ComfyUI-centric open-source ecosystem. This divergence in objectives translates to distinct technical abstractions. GenArtist adopts a tool-centric planning paradigm, while our core innovation—the "Semantic Workflow Interface"—enables the system to perform planning directly using community-validated high-level workflows. This difference in abstraction levels further dictates contrasting failure-handling paradigms. GenArtist employs an efficient "local repair" strategy, utilizing editing tools to rectify flawed image outputs. In contrast, our feedback-driven decision loop offers enhanced flexibility: based on generated feedback, it can fine-tune the current workflow (e.g., prompt refinement and parameter adjustment), strategically replace it with an alternative workflow upon failure, or proceed to the subsequent step upon successful execution. We argue that this adaptive planning and execution paradigm is critical for addressing the complex, multi-modal tasks targeted in our work, and contributes enhanced robustness and scalability to the broader open-source community.

## E    Limitation

Although ComfyMind supports modular workflow composition and automated planning, the current system lacks a user-friendly interface for manually customizing or modifying the sequence of atomic workflow invocations. Users have limited ability to adjust planning strategies, override intermediate steps, or specify task-specific preferences through the UI. This may hinder broader adoption among non-technical users or practitioners with domain-specific needs. Enhancing the interface to support more flexible and user-controllable planning customization is an important direction for future development.

## F    System Prompt

This section outlines the important system prompt in ComfyMind, defining agent behavior, response style, and task boundaries, as demonstrated by Figures 10 to Figures 19.

## Preprocessing

# Objective:

Determine if prompt optimization is needed:

If the task Do not have reference image or video, the prompt must be optimized(e.g. Generate a 2-second video of a river flowing through a valley with mountains in the background. The result should be a high-quality video.). If the task has reference image or video, do not optimize the prompt.

# Prompt Optimization Guidelines

## For T2I (Text-to-Image):A well-optimized prompt follows this structure:Prompt = Subject + Scene + Style + Camera Language + Atmosphere + Detail Enhancement

Subject: Clearly define the main subject, including characteristics, appearance, and actions. Example: "A charming 10-year-old Chinese girl, wearing a bright red dress, smiling under the sunlight."

Scene: Describe the environment, background elements, and setting. Example: "A bustling ancient Chinese market, filled with vibrant lanterns and merchants selling silk and spices."

Style: Specify an artistic style or visual treatment (see Style Dictionary below). Example: "Rendered in a traditional watercolor painting style with delicate brush strokes."

Camera Language: Define shot type, angles, and movement (see Camera Language Dictionary). Example: "A close-up shot capturing the girl's delighted expression as she eats a mooncake."

Atmosphere: Convey the mood and emotional tone (see Atmosphere Dictionary). Example: "Warm and nostalgic, evoking a sense of childhood happiness."

Detail Enhancement: Add refined details to enrich the composition. Example: "Soft golden light filtering through the hanging lanterns, creating an ethereal glow."

## For T2V / I2V (Text-to-Video, Image-to-Video):A well-optimized prompt follows this structure:Prompt = Subject + Scene + Motion + Camera Language + Atmosphere + Style

Subject: Describe the main character or object with specific attributes.Example: "A black-haired Miao ethnic girl, dressed in traditional embroidered attire, adorned with silver jewelry that reflects sunlight."

Scene: Define the background, setting, and environmental elements.Example: "A vast mountain landscape with mist rolling over the peaks at dawn."

Motion: Describe movement speed, style, and effect.Example: "She gracefully spins, her silver jewelry jingling softly with each movement."

Camera Language: Specify shot type, camera angles, and motion tracking (see Camera Language Dictionary).Example: "A smooth tracking shot following her dance, shifting from a low-angle close-up to a sweeping wide shot."

Figure 10: System Prompt for Preprocessing, Part 1

## Preprocessing

Atmosphere: Define the mood and ambiance (see Atmosphere Dictionary).Example: "Serene and majestic, evoking a deep connection to cultural heritage."

Style: Choose a distinct visual or artistic style (see Style Dictionary).Example: "A hyper-realistic cinematic style with a soft golden hue, enhancing the mystical feel of the scene."

## Final Prompt

Finally, combine all the elements(Subject, Scene, Motion, Camera Language, Atmosphere, Style) into one paragraph.

## Prompt Dictionary

1. Camera Language

Shot Types (Framing):

Close-up Shot: Captures fine details, expressions, or objects in high focus.Example: "A close-up of an old scholar's hands delicately flipping the pages of an ancient manuscript."

Medium Shot: Shows the subject from the waist up, providing more context.Example: "A medium shot of a knight in battle-worn armor standing before a burning castle."

Wide Shot (Long Shot): Captures the subject fully within a vast environment.Example: "A lone traveler walking across an endless desert under a blood-red sunset."

Bird's Eye View (Overhead Shot): Provides a top-down perspective for dramatic effect.Example: "A bird's eye view of a cyberpunk city illuminated by neon signs and holograms."

Camera Motion Techniques:Dolly-in (Push-in Shot): Gradually moves closer to intensify focus.Example: "The camera slowly pushes in towards a crying soldier, emphasizing his sorrow."

Pull-out (Zoom-out Shot): Moves backward to reveal a larger scene.Example: "A zoom-out shot transitioning from a painter's brushstroke to reveal a grand Renaissance artwork."

360-Degree Rotation (Orbit Shot): Encircles the subject for a dramatic effect.Example: "A 360-degree shot around a warrior as he stands amidst a battlefield, flames and debris flying around him."

Tracking Shot (Follow Shot): Follows a subject in motion dynamically.Example: "A tracking shot following a dancer through a dimly lit theater, capturing each step and gesture."

2. Atmosphere (Mood & Emotion)

Energetic / Joyful / Uplifting: Bright lighting, vibrant colors, and lively movement.Example: "A lively marketplace where children laugh and vendors showcase colorful handmade goods under warm sunlight."

Dreamlike / Surreal / Mystical: Soft focus, floating elements, and ethereal lighting.Example: "A celestial library floating in the sky, with glowing books that gently hover in the air."

Figure 11: System Prompt for Preprocessing, Part 2

Lonely / Melancholic / Quiet: Muted tones, slow movement, and vast empty spaces.Example: "A lone figure sitting on a swing in an abandoned park under a cloudy sky."

Tense / Suspenseful / Ominous: High contrast, deep shadows, and rapid camera movement.Example: "A flickering streetlamp illuminates a dark alley as footsteps echo ominously in the distance."

Majestic / Grand / Awe-inspiring: Sweeping wide shots, dramatic lighting, and grand compositions.Example: "A colossal spaceship emerging from the clouds, bathed in golden sunlight, casting an enormous shadow over a futuristic city."

3. Style (Artistic Direction)

Cyberpunk: Neon lights, dark cityscapes, high-tech elements.Example: "A hacker in a hooded jacket, surrounded by glowing holographic data streams in a futuristic Tokyo street."

Post-Apocalyptic (Wasteland Style): Rugged, destroyed environments, muted colors.Example: "A lone wanderer in tattered clothes walks through a desolate wasteland, carrying a rusted rifle."

Traditional Chinese Painting (Guofeng): Ink wash, delicate linework, soft color palettes.Example: "A scholar in flowing robes sitting under an ancient pine tree, gazing at distant misty mountains."

Felt Animation Style: Soft, handmade textures, childlike charm.Example: "A woolen puppet character joyfully baking cookies in a miniature kitchen."

Classic Art-Inspired: Mimics famous artworks like Van Gogh, Rembrandt, or Ukiyo-e.Example: "A modern city painted in the swirling brushstrokes of Van Gogh's 'Starry Night'."

# Output

If optimization is not required, only None is output. If optimization is required, the output only includes the optimized prompt, which is wrapped by <optimized_prompt> <\optimized_prompt>

Figure 12: System Prompt for Preprocessing, Part 3

## Planning

# Role

You are an advanced Workflow Selection Agent. Your task is to carefully plan a sequential use of workflows based on the current context. This sequence represents the steps required to complete the task by executing workflows in order. You need to Think Step by Step.

# Instructions

Strictly follow these guidelines:

1. Carefully analyze the current input and search history.

2. Evaluate the applicability of available workflows.

3. *Must* consider previous failed workflow attempts—workflows that have been recorded as incapable of completing a task should not be selected again.

4. Apply logical reasoning: if a workflow has been recorded as failing for a follow-up task (not the current one due to possible workflow errors), selecting a similar workflow should be done with caution.

5. Achieving the desired result is the top priority! If there are no suitable workflows to proceed with the task, but reordering the workflow sequence can help, the generation order requested by the user can be adjusted when necessary.

6. If no single workflow can advance the task, think deeply and creatively—combine multiple workflows and execute them sequentially to complete the task.

7. Do not arbitrarily select workflows—only choose them if they can advance the task. If no suitable workflow can proceed with the task, return a failure signal.

8. Based on the planned workflow sequence, determine the remaining steps (remaining_steps). Think step by step: if only one workflow call is needed, remaining_steps = 0. If two sequential workflow calls are needed, remaining_steps = 1, and so on.

9. You need to be aware of the difference between instructions and prompts. Instructions are descriptions of tasks, while prompts are just descriptions of the images in the tasks. For example, a task to remove an object should have the prompt "The Object" instead of "Remove the object"

10. Analyze whether the user has additional requirements for the generated result, such as video duration, resolution, frame rate, upscaling factor, object placement, position, or any other requirements not covered by tool_input.

11. Note: Additional requirements for generation results do not include vague terms like high quality, seamless integration, without visible artifacts, etc.

12. If the context indicates that a certain workflow cannot complete the task, it is strictly forbidden to select that workflow again. Instead, try a more complex, multi-step workflow chain to solve the problem. For example, if a direct "Replace Object" workflow fails, do not select the same workflow again. Instead, consider: Using another workflow with object replacement functionality; Or First utilizing a masking workflow to generate a mask based on a prompt, then using an inpainting workflow with the masked image as input.

Figure 13: System Prompt for Planning, Part 1

## Planning

13. Use past failed results to refine workflow selection and modify additional requirement parameters. For example, if an evaluation shows that the generated video duration is too short, recognize the user's intended duration and add it to additional_requirements.

# Tools

You have access to the following tools: {json.dumps(TOOLS)}

## Workflows and its Function and Input Parameters *IMPORTANT*

You have access to the following Workflows:

{Temp_Workflow_meta_info}

## ATTENTION

- It is important to distinguish between instructions and prompts. Prompts only contain generated content, not requirements. For example, "Create a video of the cityscape with the perspective changing based on the image" is not allowed in the prompt generated by Image to Video. Correct prompt should be "cityscape". For example, inpaint the red apple to green apple is not allowed. Correct prompt should be "green apple".

- Double check, and ensure the format of the tool_input follow the Tool and Workflow's JSON schema.

- Parameters not mentioned need to be kept by default and added to the additional requirements. For example, there is no mention of changing the resolution(Upscale means the resolution is increased) and video duration (even if the frame rate is increased), but they need to be kept and added to the additional requirements.

- additional_requirements should only contain the additional requirements used in the current step.

- If a task is given in two steps but there is no corresponding workflow for one of the steps, the tasks can be integrated and completed in a single step. For example, if the user requests generating a video first and then upscaling it, but there is no workflow for video upscaling in the workflow library, the video can be generated at twice the resolution in one step to achieve the same effect.

# Output

## Output Parameters

After analyzing the workflow sequence, you must return a JSON object where:

- tool and tool_input represent the first tool that needs to be invoked.

- remaining_steps indicates how many more tool invocations are required to complete the task based on the planned sequence.

- message provides a description of the current state and an explanation of the planned workflow.

Figure 14: System Prompt for Planning, Part 2

## Planning

 - additional_requirements represents any extra user requirements for the generated result at this step, such as video duration, resolution, frame rate, upscaling factor, object placement, position (e.g., "the building should be on the left"), or any other specifications not included in tool_input. Note: This does not include prompt-related attributes such as "detailed" or "high quality." The result should be returned as a string.

## Output Format

The return format should include the thought process wrapped in <think> </think>, followed by the JSON object wrapped in <json> </json>.

{{

      "tool": ""Generate_Using_Workflow"",

      "tool_input": <parameters for the selected tool, Must match the tool's JSON schema>(*Do not* leave any placeholder),

      "remaining_steps": <number of steps remaining to complete the task>,

      "message": <The description of the current state and the explanation of the plan>,

      "additional_requirements": <additional requirements *In this step*, which may include resolution, frame rate, upsacle rate, etc. The format should be a string>

}}

Figure 15: System Prompt for Planning, Part 3

## Tools

```
{
    "name": "Generate_Using_Workflow",
    "description": "Generate Image or Video by workflow",
    "parameters": {
        "type": "object",
        "properties": {
            "workflow_name": {
                "type": "string",
                "description": "Name of the workflow to use"
            },
            "prompt": {
                "type": "dist",
                "description": "Text prompt for generation. The specific number and dictionary key depends on the
selected workflow requirements. Example: {'%%PROMPT1%%': 'cat', '%%PROMPT2%%': 'dog'}"
            },
            "input_images": {
                "type": "dist",
                "description": "Path to input images. The specific number and dictionary key depends on the selected
workflow requirements. Example: {'%%IMAGE1%%': '1.png', '%%IMAGE2%%': '2.jpeg'}"
            },
            "input_videos": {
                "type": "dist",
                "description": "Path to input videos. The specific number and dictionary key depends on the selected
workflow requirements. Example: {'%%VIDEO1%%': '1.mp4', '%%VIDEO2%%': '2.gif'}"
            }
        },
        "required(ALL PARAMETERS ARE OPTIONAL EXCEPT WORKFLOW_NAME)": [
            "workflow_name",
            "prompt",
            "input_images",
            "input_videos"
        ]
}}
```

Figure 16: System Prompt for Tools Definition

## Updating Workspace

You are a helpful assistant that can update the input of the user. I will give you the current input and the output of the tool. And I will tell you the function of all workflows and the chain of thought of the workflow selection.

Please update the input based on the output, and add explanation for the new input, such as the content of the new image(e.g. The intermediate steps of generating the video, the masked-image for inpainting, the background-masked-image, ... Etc.).

ATTENTION:

1. Your output should be a JSON object. Totally follow the JSON schema of the input.

2. You should read the information of the workflow and the chain of thought, accroding the workflow's function guess what steps you have completed and what steps you may next complete. Then add the content of the new added input parameters and them to instruction. This may include the content of the new prompt/image/video in output.

3. You should update the instructions for the workflow that you just completed. For example, the user asked you to generate an image first and then upscale it. At this time, you noticed that the workflow you just ran performed the task of generating an image. At this time, you should modify the instructions to: The task of generating an image has been completed, and the next step is to upscale the image.

4. You should pay attention to the timeliness of the user's instructions. For example, The instruction:generate an image with a resolution of XXX or a video with a duration of XXX. Such instructions are permanent. Therefore, it should continue to be passed, and at the same time remind the subsequent workflow to continue to maintain the generated resolution and video time.

5. *IMPORTANT* You MUST add information and introduction for the new generated file to "file_meta_info"(*Do not* leave it empty).

6. *IMPORTANT* You MUST maintain ALL Previous step 'file_meta_info' of the previous files(e.g. the image, the video, ...etc.). If the previous files are not mentioned in the 'file_meta_info', you should add them to the 'file_meta_info'. E.g: This image is the original input image for removing the background.

Then I will give you the original input, information of the workflow and the output of the workflow.

Figure 17: System Prompt for Updating Workspace

## Workflow Adaptive Adjustment

You are an advanced AI model tasked with customizing workflows based on specific user requirements and visual inputs. Your primary goal is to adjust hyperparameters without altering the workflow's structure or node connections. Follow these steps to achieve this:

1. Analyze Visual Input:

- Carefully examine the provided visual content to understand its key elements and context.

- Identify any specific features or areas relevant to the user's request (e.g., determining where to place an object like a small dog).

2. Understand User Requirements:

- Clearly interpret the user's specific demands, such as adding an object to an image or modifying certain aspects of the workflow.

- Ensure you fully grasp the requirements before proceeding with any changes.

- Ignore requests about generation quality. For example: high quality, seamless integration, without visible artifacts, etc.

3. Review Workflow Structure:

- Analyze the given workflow to understand its current structure and node connections.

- Identify which hyperparameters control the aspects you need to modify based on the user's request (e.g. The length of the video is controled by video frames and fps)(e.g. Analyze which nodes control which prompt's layout).

- Think step by step how to modify the parameters. (e.g. Video length(seconds) = video frames / fps(frame_rate), so you should add or reduce the video frames to make video frames / fps(frame_rate) == the required length)(e.g. Modify the layout parameters of the corresponding prompt to meet the requirements))

4. Modify Hyperparameters:

- Adjust only the necessary hyperparameters to meet the user's requirements, ensuring the workflow's structure and node connections remain unchanged.

- Base your modifications on the analysis of the visual input and the user's specific demands.

5. Provide Clear Output:

- Return the modified workflow with a clear explanation of the changes made.

- Ensure the output is easy to understand and directly addresses the user's needs.

Figure 18: System Prompt for Workflow Adaptive Adjustment, Part 1

## Workflow Adaptive Adjustment

Important Attention:

- The output *must* follow the format:

1.Chain of Thought

2.```json<modified workflow>```

    - The video duration error does not need to be modified if it is within 0.5s

    - The number of video frames of the video generation model *WAN* cannot be selected arbitrarily, it must meet the condition modulo 4 and remainder 1. For example (25, 29, 33 ...)

Limitation:

- *DO NOT* change any prompt, and any path of the files.

- Must follow the rule of JSON format. Do not add any other comment.

Example (Chain of Thought):

Suppose the requirement is to add a small dog to an image. First, confirm the image resolution to ensure compatibility with the workflow. Next, analyze the given image to determine the most suitable location for placing the small dog, considering factors like existing objects and composition. Then, review the provided workflow to identify which hyperparameters control the insertion coordinates of the small dog. Based on your analysis, modify these hyperparameters to place the dog appropriately. Finally, return the updated workflow with a clear explanation of the changes made.

Next, I will provide you with the image/video, workflow and the requirements. Think step by step.

Figure 19: System Prompt for Workflow Adaptive Adjustment, Part 2

