# OpenReview forum: "ComfyMind: Toward General-Purpose Generation via Tree-Based Planning and Reactive Feedback"
_NeurIPS.cc/2025/Conference — NeurIPS 2025 poster_

### Official Review · Reviewer_Hh8S · 2025-07-01

**Clarity:** 3
**Significance:** 3
**Originality:** 2
**Rating:** 4
**Confidence:** 3

**Summary:**

This work proposes ComfyMind, which is a planning system built on top of ComfyUI, a node-based execution system for image/video generation. ComfyMind contains two main components: 1) a semantic interface that allows natural language inputs and 2) a planning algorithm with localized feedback that searches for stable and high-quality workflows. ComfyMind has been evaluated on open-source benchmarks, and results showed that it consistently outperforms existing open-source baselines.

**Questions:**

n/a

**Ethical Concerns:**

["NO or VERY MINOR ethics concerns only"]

**Final Justification:**

The additional experiments show good stability and efficiency of the algorithm. I will keep my score.

**Quality:**

2

**Strengths And Weaknesses:**

Strengths:
Clarity: The work is presented clearly, with well-defined motivations and a straightforward, easy-to-follow pipeline. Detailed comparisons with prior works further enhance understanding.
Significance: The proposed algorithms have been proven to be useful in three sets of experiments.

Weaknesses:
While the search tree planning algorithm is described at a high level, and its efficacy is justified through experimental results and ablation studies, a more detailed quantitative analysis would strengthen the work. For instance, the claim that "This allows the system to..., ...lead to strategy oscillations and failure to converge" could benefit from direct quantitative comparisons (e.g., efficiency, stability) between the proposed method and baselines like ReAct. Presenting experiments with more detailed ananysis, rather than the final metric, would be helpful.

---

> ### Author Rebuttal · Authors · 2025-07-31
>
> We sincerely thank you for your thorough review and constructive suggestions. In response to your comment, we have added a more direct and detailed quantitative analysis to support our views.
>
> **> W1.**
>
> > The claim that “This allows the system to… avoid strategy oscillations” could benefit from direct quantitative comparisons (e.g., efficiency, stability) between the proposed method and baselines like ReAct.
>
> *Answer:*
>
> To address your concern regarding the need for more detailed quantitative analysis, we conducted comprehensive comparisons with representative planning strategies—ReAct and Planning-Act—on the ComfyBench benchmark. The results are presented below:
>
> | Method | Resolve Rate \% ↑ | Repetition Rate \% ↓ | Time (Seconds) ↓ |
> |--------|-------------|----------------|----------------|
> | ReAct | 54.0 | 34.59 | 257.43 |
> | Planning-Act | 66.0 | 28.24 | 279.79 |
> | Ours | **83.0** | **8.70** | **225.12** |
>
> We evaluate three core metrics: **Resolve Rate**, **Repetition Rate**, and **Average Time per Task**, which respectively reflect a system’s **problem-solving capability**, **stability of exploration**, and **efficiency**.
>
> **Resolve Rate** measures the success rate in solving compositional tasks, directly demonstrating the planner’s ability to construct effective multi-step generations.
>
> **Repetition Rate** captures the frequency with which atomic workflows are redundantly selected within a single task. High repetition suggests issues such as oscillations or failure to adapt contextually. Our method significantly reduces this rate, indicating more stable and purposeful search behavior.
>
> **Time** represents the average execution time per task. Our approach benefits from broader exploration and avoids falling into repetitive local patterns, allowing it to solve problems earlier and more efficiently.
>
> For **ReAct**, its higher repetition rate (34.59) and lower success rate (54.0) suggest wasted exploration. Although ReAct executes actions in a reactive manner based on current observations, it often lacks global planning and stable intermediate decision nodes. This leads to repeated trial-and-error behavior, resulting in inefficient progress and strategy oscillations. The relatively high average task time (257.43s) further indicates that steps is not used efficiently, and the system spends more time cycling through ineffective action sequences.
>
> For **Planning-Act**, this is a canonical method widely adopted in multi-agent systems. It follows a structure where the agent performs full planning at the root node and then recursively executes subtasks. However, this architecture has two major drawbacks:
> 1. It lacks real-time adaptability: though it can re-plan after errors, its plan fixes once a plan-act cycle starts, with no way to adjust mid-execution based on results—e.g., it can’t dynamically add an image enhancement workflow if quality is too low mid-step. This rigidity also limits exploring similar-function workflow branches, hindering optimal solutions and contributing to its lower Resolve Rate (66.0%).
> 2. Due to the absence of stable intermediate decision nodes, it frequently needs to reselect workflows required for intermediate steps, leading to significant redundancy and inefficiency. These limitations are reflected in its high Repetition Rate (28.24), and the longest average Time per Task (279.79s) among all compared methods.
>
> In contrast, **our method** achieves the best performance across all three dimensions: higher success, lower redundancy, and faster convergence.
>
> This analysis provides concrete empirical evidence supporting our claim that the proposed planning mechanism mitigates strategy oscillations and enhances convergence efficiency, outperforming existing baselines both in effectiveness and in behavioral stability.

---

> > ### Comment · Reviewer_Hh8S · 2025-08-03
> > **response to rebuttal**
> >
> > I would like to thank the authors for the response. The additional experiments show good stability and efficiency of the algorithm. I will keep my score.

---

### Official Review · Reviewer_yctv · 2025-07-02

**Clarity:** 4
**Significance:** 4
**Originality:** 4
**Rating:** 5
**Confidence:** 3

**Summary:**

This paper introduces ComfyMind, a collaborative AI system built on the ComfyUI platform, designed for robust and scalable general-purpose generation. Addressing the fragility of existing open-source frameworks in complex real-world applications, ComfyMind introduces two core innovations: a Semantic Workflow Interface (SWI), which abstracts low-level node graphs into callable functional modules described in natural language, enabling high-level composition and reducing structural errors; and a Search Tree Planning mechanism with localized feedback execution, which models generation as a hierarchical decision process, allowing adaptive correction at each stage. Evaluated on three public benchmarks (ComfyBench, GenEval, and Reason-Edit) spanning generation, editing, and reasoning, ComfyMind consistently outperforms existing open-source baselines and achieves performance comparable to GPT-Image-1, paving a promising path for open-source general-purpose generative AI systems.

**Questions:**

1. Within the search tree planning, is MCTS (Monte Carlo Tree Search) or another specific search algorithm used to guide node selection and path exploration? If not, how does the agent effectively avoid local optima or inefficient exploration?

2. The localized feedback relies on a VLM for parsing and annotating generated content. The VLM's own performance and its ability to understand complex visual details could become a bottleneck for feedback quality and overall system robustness.

**Ethical Concerns:**

["NO or VERY MINOR ethics concerns only"]

**Final Justification:**

Overall, the responses are detailed, well-motivated, and supported by new empirical evidence. They strengthen the clarity, technical rigor, and extensibility of the work. I am satisfied that the authors have adequately addressed the concerns, and the revised version will be significantly improved.

**Limitations:**

See weakness and questions.

**Paper Formatting Concerns:**

Overall, the responses are detailed, well-motivated, and supported by new empirical evidence. They strengthen the clarity, technical rigor, and extensibility of the work. I am satisfied that the authors have adequately addressed the concerns, and the revised version will be significantly improved.

**Quality:**

4

**Strengths And Weaknesses:**

Strengths:

1. Innovative High-Level Semantic Abstraction: The Semantic Workflow Interface (SWI) is a key highlight, abstracting ComfyUI's low-level node operations into natural-language function modules. This significantly simplifies workflow construction, reduces syntax errors, and enhances the language model's ability to understand and compose complex tasks.

2. Robust Localized Feedback Mechanism: The Search Tree Planning with Local Feedback Execution is highly effective. By modeling workflow construction as a hierarchical decision process and triggering local corrections upon execution failures, it avoids global rollbacks and redundant computations, greatly enhancing system robustness in complex tasks.

3. Broad Generality and Excellent Performance: ComfyMind demonstrates strong performance across three diverse benchmarks (ComfyBench, GenEval, Reason-Edit) for generation, editing, and reasoning. It not only significantly outperforms open-source baselines like ComfyAgent but also matches or even surpasses the closed-source GPT-Image-1 in some aspects, proving its powerful general-purpose generation capabilities.

Weakness

Localized feedback is crucial for ComfyMind's robustness, but the paper does not quantify how this feedback is implemented. For instance, the granularity of VLM parsing and annotation, the specific format of failure signals, and how parent nodes revise their planning based on this feedback are not detailed.

---

> ### Author Rebuttal · Authors · 2025-07-31
>
> We sincerely thank you for your detailed and insightful comments. Below, we provide point-by-point responses to your concerns and suggestions.
>
> **> W1.**
>
> > The paper does not quantify how the feedback is implemented… the granularity of VLM parsing, failure signal formats, and how parent nodes revise plans are not described.
>
> *Answer:*
>
> To directly address your concern regarding the robustness of localized feedback, we conducted additional experiments on ComfyBench, analyzing how different levels of feedback granularity affect task success rates.
>
> | Method | Vanilla \% ↑ | Complex \% ↑ | Creative \% ↑ | Total \% ↑ |
> |--------|---------|---------|----------|-------|
> | Judgment | 89.0 | 58.33 | 47.5 | 71.5 |
> | Judgment+Analysis | 91.0 | 66.67 | 55.0 | 76.5 |
> | Judgment+Analysis+Description(Ours) | **92.0** | **85.0** | **57.5** | **83.0** |
>
> Here,
> - Judgment refers to feedback containing only a binary success/failure signal.
> - Analysis adds explanations of the outcome, specifying the reasons for success or failure.
> - Description includes more detailed image-grounded feedback, such as what has been completed, visual observations, and suggestions for correcting the current plan or direction.
>
> These results demonstrate the effectiveness of our localized feedback design. Increasing feedback granularity provides richer information for the planning agent, improving its ability to make informed decisions. This benefit is further amplified by our search tree design, which prevents the context from becoming too long at each planning node—ensuring that even fine-grained feedback remains effective without being diluted.
>
> As for the format of failure signals and how parent nodes revise plans:
> - Failure signals are returned to the parent node in a structured JSON format.
> - The plan revision mechanism works by injecting the feedback content into the parent node’s context. The planning agent then performs reasoning based on this context, which includes the description of past failures, previously chosen workflows, and the current input. It avoids paths known to be incorrect or unsuccessful and selects the next best atomic workflow based on this updated information.
>
>
>
> **> Q1.**
>
> > Is MCTS (Monte Carlo Tree Search) or another specific search algorithm used? If not, how does the agent avoid local optima or inefficient exploration?
>
> *Answer:*
>
> Thank you for this insightful question. We do not use MCTS or any traditional search algorithm. Instead, our system is driven by LLM-based heuristic search, leveraging the strong reasoning and planning capabilities of LLMs.
>
> These capabilities enable the model to perform informed and adaptive path selection, effectively avoiding local optima without the need for additional tree search mechanisms. When the agent encounters a dead-end or inefficient trajectory, it can proactively exit the path and re-plan. We also set an upper limit of 5 candidate paths per planning agent, which balances sufficient exploration of the solution space while preventing local optimality.
>
> Another reason we deliberately avoid using MCTS or reinforcement learning (e.g., GRPO) is to preserve generality and ease of use. Training in specific domain may improve performance, but reduce generalization and hinder the system's ability to incorporate stronger workflows or components in the future. Our design emphasizes plug-and-play modularity and general-purpose applicability across domains.
>
> **> Q2.**
>
> > The VLM's own performance in understanding complex visual details could become a bottleneck for the feedback quality and system robustness.
>
> *Answer:*
> We fully agree with this insightful observation. The performance of the evaluation VLM is indeed a critical factor, and its potential for misjudgment in complex scenes is a known limitation we have actively considered in our design.
>
> In our current implementation, we already mitigate this to some extent by using structured(Judgment, Analysis, Description) and targeted queries for evaluation, which constrains the VLM's task and reduces ambiguity. However, to further enhance robustness, our modular framework is designed for a straightforward extension to a multi-granularity feedback mechanism. This involves a hierarchy of checks: a high-level VLM for overall semantic coherence, combined with more specialized models (e.g., an object detector or attribute classifier) for fine-grained verification. This approach reduces the risk of single-point failure by cross-validating feedback. We will add a detailed discussion of this enhancement strategy to the appendix of the final version.
>
> We also concur that as foundational VLMs continue to rapidly advance, this bottleneck will naturally diminish. Our plug-and-play architecture is perfectly positioned to seamlessly benefit from these future improvements.

---

> > ### Comment · Reviewer_yctv · 2025-08-04
> >
> > Thanks for your detailed response. I will keep my score.

---

### Official Review · Reviewer_j82d · 2025-07-06

**Clarity:** 4
**Significance:** 2
**Originality:** 2
**Rating:** 4
**Confidence:** 5

**Summary:**

This paper introduces ComfyMind, a system that optimizes prompts through reactive feedback and leverages downstream models to support a wide range of visual generation tasks. Specifically, the proposed Semantic Workflow Interface (SWI) abstracts low-level nodes into callable modules, facilitating high-level composition across diverse tasks. Extensive experiments demonstrate the advantages and effectiveness of ComfyMind in visual generation, editing, and reasoning.

**Questions:**

+ Apart from the final results, an analysis of the LLM planning process should be conducted. How robust is this process? If the LLM makes incorrect decisions or generates flawed prompts, will the errors propagate to the final performance? How can this potential issue be mitigated? What is the bottleneck of the proposed pipeline — the LLM or the downstream visual generation models?
+ In the appendix, instead of only showing the final visualizations, the entire LLM reasoning process should be presented. Failure cases should also be discussed to inform future research improvements.
+ It is already known that prompt rewriting can significantly boost GenEval performance. In Table 2, do the baseline models also use rewritten prompts? If not, is this a fair comparison?
+ Since ComfyMind requires LLM planning and iterative feedback, how does its efficiency compare to each of the target baselines? Does the performance gain justify the additional time overhead?

**Ethical Concerns:**

["NO or VERY MINOR ethics concerns only"]

**Limitations:**

yes

**Quality:**

3

**Strengths And Weaknesses:**

**Main Strengths**
+ The paper is well-written and easy to follow.
+ It provides numerous qualitative examples and system prompts, which support the reproducibility of the proposed system.
+ The proposed pipeline is model-agnostic, allowing newer downstream visual generation models to benefit further.

**Main Weaknesses**
+ Although the performance is promising, ComfyMind appears to be a combination of LLM agents [1, 2] and prompt rewriting techniques [4, 5, 6], which are already well-known to enhance visual generation.

[1] DiffAgent: Fast and Accurate Text-to-Image API Selection with Large Language Model

[2] GenArtist: Multimodal LLM as an Agent for Unified Image Generation and Editing

[3] Improving Image Generation with Better Captions

[4] Tailored Visions: Enhancing Text-to-Image Generation with Personalized Prompt Rewriting

[5] Dynamic Prompt Optimizing for Text-to-Image Generation

[6] Prompt Adaptation as a Dynamic Complement in Generative AI Systems

---

> ### Author Rebuttal · Authors · 2025-07-31
>
> We sincerely thank you for your clear, detailed, and thought-provoking comments. Below, we address each of your concerns and questions point by point.
>
> **> W1.**
>
> > ComfyMind appears to be a combination of LLM agents and prompt rewriting, which are known to enhance visual generation.
>
> *Answer:*
>
> We sincerely thank the reviewer for astutely pointing out the connection between our research and the two important research directions of LLM agents and prompt optimization. We acknowledge that these studies have laid an important foundation for our work. However, our research is not a simple combination of these two.
>
> The originality of our work lies in providing a complete, reliable, and adaptive framework and technical pathway based on ComfyUI. The Semantic Workflow Interface addresses the frequent omissions and syntax errors in workflow construction that plague other methods, enabling agents to focus on the ability to compose and call at a high level. Based on SWI, Search Tree Planning allows for exploring more and better solutions while maintaining efficiency. Finally, Local Feedback Execution enables the system to maintain generation quality while supporting fine-grained adaptive decision-making: for instance, when the current path is ineffective, it can strategically replace the entire workflow; when the problem is complex, it will continue to compose new capabilities; and when there are only minor deviations, it will optimize its prompts or parameters. In conclusion, our work is not a simple superposition of existing technologies, but rather provides a promising approach for the open-source generative ecosystem through interlocking designs.
>
> **> Q1.**
>
> > Apart from the final results, an analysis of the LLM planning process should be conducted. How robust is this process? If the LLM makes incorrect decisions or generates flawed prompts, will the errors propagate to the final performance? How can this potential issue be mitigated? What is the bottleneck of the proposed pipeline — the LLM or the downstream visual generation models?
>
> *Answer:*
>
> To directly address your concerns about the robustness of the LLM planning process, we conducted comprehensive comparisons against representative planning strategies — ReAct and Planning-Act — on the ComfyBench benchmark. Planning-Act is a canonical method widely adopted in multi-agent systems, which performs full planning at the root node and then recursively executes subtasks in a top-down manner. The results are as follows:
>
> | Method | Resolve Rate \% ↑ | Repetition Rate \% ↓ |
> |--------|-------------|----------------|
> | ReAct | 54.0 | 34.59 |
> | Planning-Act | 66.0 | 28.24 |
> | Ours | **83.0** | **8.70** |
>
> Here, Repetition Rate reflects how often the planner redundantly selects the same atomic workflow. Compared to ReAct and Planning-Act, our method achieves the lowest repetition rate, indicating fewer oscillations and better learning from past failures. The higher resolve rate also demonstrates broader exploration and stronger robustness of the planning process.
>
> We acknowledge that LLMs can make flawed decisions or generate incorrect prompts. Our system is designed to significantly mitigate their downstream effects. Syntax errors and atomic workflow failures are typically detected early, prompting the planning agent to replan or backtrack. For other errors (e.g., low-quality generations or flawed prompts), the evaluation agent identifies the issue and triggers Local Feedback Execution. When the rollback reaches the erroneous planning node, the system re-plans accordingly. In this way, while errors may increase generation time or path length, they are effectively contained and do not propagate to the final output.
>
> As for the main bottleneck of our pipeline, we believe it lies in the downstream visual generation models. In most failure cases, the LLM agent generates reasonable plans, but the outputs fail due to insufficient capability or instability of the generative models.
>
> **> Q2.**
>
> > In the appendix, instead of only showing the final visualizations, the entire LLM reasoning process should be presented. Failure cases should also be discussed.
>
> *Answer:*
>
> Thank you for the valuable suggestion. We fully agree that providing complete processing traces and failure cases, which is critical for transparency and deeper understanding. We have already collected a set of representative examples, covering both successful reasoning steps and typical failure cases. Due to policy constraints, we are currently unable to include associated images or PDFs. However, we firmly commit to adding detailed reasoning traces and error case discussions in the appendix of the revised version.
>
> **> Q3.**
>
> > In Table 2, do the baseline models also use rewritten prompts? If not, is this a fair comparison?
>
> *Answer:*
>
> In Table 2, most baseline models were indeed evaluated using their default inputs, without explicit prompt rewriting. To ensure a fair comparison, the Prompt Preprocessing step—an explicit external rewriting module—was disabled during our experiments, and the original prompts were directly provided to the system.
>
> However, within ComfyMind, prompt rewriting can occur as a dynamic internal step—either during planning (e.g., to adjust prompts dynamically when generation quality is insufficient) or through workflows explicitly designed for prompt optimization. This internal rewriting is considered part of ComfyMind’s overall adaptive capability rather than external preprocessing.
>
> We will gladly add a more explicit ablation study in our revised manuscript, ensuring an absolutely fair comparison by enabling comparable rewriting mechanisms for baseline methods
>
> **> Q4.**
>
> > Since ComfyMind requires LLM planning and iterative feedback, how does its efficiency compare to each of the target baselines? Does the performance gain justify the additional time overhead?
>
> *Answer:*
>
> We present two sets of comparisons to address your question about efficiency and performance trade-offs:
>
> | Model          | **Time(Seconds) ↓** | **Overall Score ↑** | Single Obj.↑| Two Obj.↑| Counting↑| Colors↑| Position↑| Attr. Binding↑|
> |----------------|---------------------|---------------------|-------------|----------|----------|--------|----------|---------------|
> | SD3.5-L        | **47.06**               | 0.71                | 0.98        | 0.89     | 0.73     | 0.83   | 0.34     | 0.47          |
> | FLUX.1 Dev     | 56.51               | 0.66                | 0.98        | 0.81     | 0.74     | 0.79   | 0.22     | 0.45          |
> | Ours           | 119.94              | **0.90**                | **1.00**        | **1.00**     | **1.00**     | **0.97**   | **0.62**     | **0.80**          |
>
>
> Although our system incurs additional time due to planning, evaluation, and possible regeneration, it achieves significantly higher overall scores. The performance gain justifies the roughly 2x increase in runtime, demonstrating that our system pushes the boundary of what current models can accomplish.
>
> | Method       | Time(Seconds) ↓ | Resolve Rate \% ↑ |
> | ------------ | -------------- | -------------- |
> | ReAct        | 257.43         |  54.0 |
> | Planning-Act | 279.79         |  66.0 |
> | Ours         | **225.12**     |  **83.0** |
>
> In broader multi-step tasks, our approach not only solves more tasks but also completes them more efficiently than other agent-based methods.
>
> Taken together, these two sets of results show that ComfyMind introduces a moderate time overhead compared to baselines but delivers substantially better performance, validating its design and effectiveness.

---

### Official Review · Reviewer_xjBj · 2025-07-06

**Clarity:** 3
**Significance:** 2
**Originality:** 2
**Rating:** 4
**Confidence:** 4

**Summary:**

This paper presents ComfyMind, a system for image generation and editing from complex text prompts by coordinating various foundation models as tools. The key idea is to utilize VLM to construct a workflow tree and call APIs from a tool library with feedback. Experiments on GenEval show promising results.

**Questions:**

What is the inference latency of the proposed method? Can authors make the comparison to the compared methods?

**Ethical Concerns:**

["NO or VERY MINOR ethics concerns only"]

**Final Justification:**

I have carefully read the authors' rebuttal and other reviews. I think this paper has its contributions and can be accepted, but the novelty is somewhat limited to me. Hence, I raise my original score to borderline accept, as I think this paper can be accepted.

**Limitations:**

The paper has discussed the limitations.

**Quality:**

3

**Strengths And Weaknesses:**

## Strenth
- ComfyMind is an agent-based system that is training-free and easy to use.
- The paper is clearly written, and the results look promising.

## Weakness
- **Major concern:** My major concern lies in the technical novelty. A closely related work, [GenArtist](https://openreview.net/pdf?id=Ur00BNk1v2) [1], proposes a similar approach and is not cited or discussed. To be specific, GenArtist also proposes to use VLM to decompose tasks into a tree-based tool execution flow that is executed with feedback control. GenArtist also supports various tasks like image generation and editing from language descriptions. While some details like the structure of every tree node are different, and ComfyMind supports more tasks by using more agents and more advanced foundation models, the core idea of ComfyMind is basically the same as GenArtist. Please clarify it and do a thorough comparison.
- Typically, such agent-based image edition and generation systems are prone to the quality of the first generated image or video. The quality might not be sufficiently good, while the evaluation might still be processed as good, since the overall quality is not bad. One way to address this might be executing image generation path several times and pick the best one, but this might be time-consuming.
- Besides the benchmarks used in this paper, please try to conduct experiments on T2I-CompBench [2] and DreamBench++ [3], and make a comparison with CompfyMind. These are good benchmarks for demonstrating the generation capability and will enrich the solidity. Note that you should try to use the same agents to make the comparison fair.


[1] Wang, Zhenyu, Aoxue Li, Zhenguo Li, and Xihui Liu. "Genartist: Multimodal llm as an agent for unified image generation and editing." Advances in Neural Information Processing Systems 37 (2024): 128374-128395. \
[2] Huang, Kaiyi, Kaiyue Sun, Enze Xie, Zhenguo Li, and Xihui Liu. "T2i-compbench: A comprehensive benchmark for open-world compositional text-to-image generation." Advances in Neural Information Processing Systems 36 (2023): 78723-78747.\
[3] Peng, Yuang, Yuxin Cui, Haomiao Tang, Zekun Qi, Runpei Dong, Jing Bai, Chunrui Han, Zheng Ge, Xiangyu Zhang, and Shu-Tao Xia. "Dreambench++: A human-aligned benchmark for personalized image generation." The Thirteenth International Conference on Learning Representations (2025).

---

> ### Author Rebuttal · Authors · 2025-07-31
>
> We sincerely thank Reviewer xjBj for the thoughtful review and constructive suggestions.
>
> **> W1.**
>
> > A closely related work, GenArtist, proposes a similar approach… Please clarify it and do a thorough comparison.
>
> *Answer:*
>
> We sincerely thank you for pointing out the important work of GenArtist. GenArtist has achieved remarkable success in automating complex visual generation using large models. Our work and GenArtist both stem from a similar vision—to tackle the challenge of complex instructions through a modular paradigm. We sincerely apologize for this oversight in citation and solemnly promise to add detailed comparative experiments in the revised version. The goals of our work and GenArtist differ: GenArtist focuses on image-related tasks, whereas our work aims to provide **a scalable and intelligent framework** for the more general-purpose, ComfyUI-centric open-source ecosystem.
>
> This difference in goals leads to different technical abstractions. GenArtist plans with **tools**, whereas our core innovation, the **"Semantic Workflow Interface"**, enables our system to directly plan with community-validated high-level **workflows**. This difference in abstraction level also dictates our different paradigms for handling failure. GenArtist employs an efficient "local repair" strategy, using editing tools to patch flawed images. In contrast, our feedback-based decision loop is more flexible: based on the feedback, it can **fine-tune** the current workflow (e.g., by modifying the prompt and parameters), **strategically replace** it with a new one upon failure, or **continue** to the next step upon success.
>
> We believe this adaptive planning and execution paradigm is crucial for the complex, multi-modal tasks we target, such as video generation, and brings greater robustness and scalability to the entire open-source community. Thank you again for your valuable feedback.
>
>
> **> W2.**
>
> > Agent-based systems are prone to the quality of the first generated image… Evaluation might still be rated as good despite poor initial quality… One way to address this might be executing the image generation path several times and picking the best one.
>
> *Answer:*
>
> We sincerely thank the reviewer for this valuable suggestion and completely agree with your point: selecting a high-quality initial generation is crucial for improving the quality of the final work. We agree with and can easily implement the solution you proposed within our framework. Specifically, our system can generate a set of candidates and then either utilize the evaluation capabilities of VLM to automatically select the best starting point, or introduce human-in-the-loop interaction to allow the user to make a personal selection. Regarding the time-consumption issue you mentioned, we plan to adopt a parallel generation strategy. Since multiple candidates can be generated simultaneously, this approach would not significantly increase the overall waiting time.  We will further integrate these strategies and will discuss them in future.
>
>
> **> W3.**
>
> > Please conduct experiments on T2I-CompBench and DreamBench++ to enrich the solidity of evaluation.
>
> *Answer:*
>
> We sincerely appreciate your suggestion regarding enhancing the robustness of our evaluation. Following your advice, we have conducted detailed experiments on T2I-CompBench and DreamBench++ while keeping the Agent component of our system unchanged.
>
> Due to time constraints, we performed experiments on the first three categories of T2I-CompBench, with the results shown below:
>
> | Method       | Color ↑   | Shape ↑ | Texture ↑ |
> | ------------ | -------- | ------ | ------ |
> | DALL-E 3     | 0.7785   | 0.6205 | 0.7036 |
> | Janus-Pro-7B | 0.6359   | 0.3528 | 0.4936 |
> | FLUX.1       | 0.7407   | 0.5718 | 0.6922 |
> | GenArtist    | 0.8482   | **0.6948** | 0.7709 |
> | Ours         | **0.8825**   | 0.6825 | **0.7866** |
>
> The experiments demonstrate that our method outperforms other approaches on T2I-CompBench. Compared to GenArtist, our method achieves better performance in the Color and Texture categories, and competitive, closely matched performance in the Shape category. These results further validate the effectiveness of our method in this text-to-image task.
>
>
> | Method          | Concept Preservation ↑  | Prompt Following ↑  |
> | --------------- | -------------------- | ---------------- |
> | BLIP-Diffusion  | 0.547                | 0.495            |
> | Emu2            | 0.528                | 0.689            |
> | IP-Adapter-Plus | 0.833                | 0.413            |
> | GenArtist       | 0.848                | 0.603            |
> | GenArtist (more tools) |0.852	         | 0.753            |
> | Ours            | **0.857**            | **0.771**        |
>
> Additionally, the table above presents the performance of our method on DreamBench++. Our method outperforms other approaches, including GenArtist and its tool-extended version, in both concept preservation and prompt following. This further illustrates the effectiveness of our method in image customization tasks.
>
> We believe this strong performance, particularly on these complex compositional and personalization benchmarks, directly stems from our system's core design. The ability to compose high-level workflows via the SWI and to adaptively select strategies allows ComfyMind to handle these challenging tasks more effectively than approaches that rely on lower-level tool planning. We will integrate these new results and corresponding analyses into the final version of our paper to substantially strengthen our evaluation section. Thank you again for this valuable guidance.
>
>
> **> Q1.**
>
> > What is the inference latency of the proposed method? Please compare with other methods.
>
> *Answer:*
>
> To address your question regarding inference latency, we have conducted detailed experiments on the ComfyBench benchmark. On Geneval, we compared the inference latency and performance of our method with two atomic workflows (FLUX.1 and SD3.5-L) that our atomic workflow is based on. The results are shown in the table below.
>
> | Model          | **Time(Seconds) ↓** | **Overall ↑** | Single Obj. ↑ | Two Obj. ↑ | Counting ↑ | Colors ↑ | Position ↑ | Attr. Binding ↑ |
> |----------------|---------------------|---------------------|-------------|----------|----------|--------|----------|---------------|
> | SD3.5-L        | **47.06**               | 0.71                | 0.98        | 0.89     | 0.73     | 0.83   | 0.34     | 0.47          |
> | FLUX.1 Dev     | 56.51               | 0.66                | 0.98        | 0.81     | 0.74     | 0.79   | 0.22     | 0.45          |
> | Ours           | 119.94              | **0.90**                | **1.00**        | **1.00**     | **1.00**     | **0.97**   | **0.62**     | **0.80**          |
>
> As can be seen, although our system requires additional time for planning, evaluation, and refinement, the increase in latency is not excessive, remaining approximately twice that of the atomic workflows. This reflects that our system can quickly converge to high-quality generation without getting stuck in oscillating and repeated attempts. Furthermore, our system achieves significantly higher performance than the underlying atomic workflows on Geneval, confirming that the additional inference time is worthwhile.
> In addition, we conducted detailed experiments on ComfyBench, which covers a broader range of task types, comparing the inference latency and performance of our method with other common planning approaches (such as ReAct and Planning-Act). Here, Planning-Act refers to the approach of first conducting global planning and then recursively executing subtasks.
>
> | Method       | Time(Seconds) ↓ | Resolve Rate \% ↑ |
> | ------------ | -------------- | -------------- |
> | ReAct        | 257.43         |  54.0 |
> | Planning-Act | 279.79         |  66.0 |
> | Ours         | **225.12**     |  **83.0** |
>
> The experiments show that our method achieves the shortest time on ComfyBench while also attaining the highest task resolution rate, demonstrating the excellent capability of our collaborative system.

---

> > ### Comment · Reviewer_xjBj · 2025-08-05
> >
> > Thank you for the detailed response, which addresses most of my concerns. Please put the new results in the revised paper so that this paper can be more solid.

---

### Decision · Program_Chairs · 2025-09-17

**Decision:**

Accept (poster)

**Comment:**

The paper received four expert reviews. The authors provided a rebuttal that attempted to address the concerns raised in the reviews. The reviewers read the rebuttal and engaged with the authors.  The reviewers unanimously like the paper and recommended accept. The area chair agreed with the recommendation and decided to accept the paper. Congratulations! Please see the reviews for feedback on the paper to revise the final version of your paper and include any items promised in your rebuttal.